# A c-di-GMP signaling module controls responses to iron in *Pseudomonas aeruginosa*

Xueliang Zhan[1,5], Kuo Zhang[2,5], Chenchen Wang [2,5], Qiao Fan[1], Xiujia Tang[3], Xi Zhang[1], Ke Wang[3], Yang Fu[2] & Haihua Liang [2,4] ✉

Cyclic dimeric guanosine monophosphate (c-di-GMP) serves as a bacterial second messenger that modulates various processes including biofilm formation, motility, and host-microbe symbiosis. Numerous studies have conducted comprehensive analysis of c-di-GMP. However, the mechanisms by which certain environmental signals such as iron control intracellular c-di-GMP levels are unclear. Here, we show that iron regulates c-di-GMP levels in *Pseudomonas aeruginosa* by modulating the interaction between an iron-sensing protein, IsmP, and a diguanylate cyclase, ImcA. Binding of iron to the CHASE4 domain of IsmP inhibits the IsmP-ImcA interaction, which leads to increased c-di-GMP synthesis by ImcA, thus promoting biofilm formation and reducing bacterial motility. Structural characterization of the apo-CHASE4 domain and its binding to iron allows us to pinpoint residues defining its specificity. In addition, the cryo-electron microscopy structure of ImcA in complex with a c-di-GMP analog (GMPCPP) suggests a unique conformation in which the compound binds to the catalytic pockets and to the membrane-proximal side located at the cytoplasm. Thus, our results indicate that a CHASE4 domain directly senses iron and modulates the crosstalk between c-di-GMP metabolic enzymes.

In response to distinct environmental stresses, bacterial cell survival is facilitated by the formation of biofilms[1,2]. Cyclic dimeric GMP (c-di-GMP), a second messenger in bacteria, is the predominant regulator of biofilm formation and motility[3]. Previous research has demonstrated that bacterial motility is partially regulated by different concentrations of c-di-GMP. For instance, higher/lower concentrations of c-di-GMP promote sessility/motility, and c-di-GMP-mediated processes are related to infection and host-microbe symbiosis[4]. The synthesis and degradation of c-di-GMP are mainly associated with two unique enzymes called diguanylate cyclase (DGC) and phosphodiesterase (PDE)[3]. Numerous studies have been reported that bacteria possess multiple c-di-GMP-related enzymes, and distinct phenotypes,

including biofilm dispersal and motility are controlled by their corresponding DGCs and/or PDEs[5]. For instance, the cellulose biosynthesis, secretion, and modification processes in *Escherichia coli* K-12 are dependent on DgcC[6]. In addition, the synthesis of extracellular polysaccharides and intracellular c-di-GMP levels in *Pseudomonas aeruginosa* PAO1 are mediated by WspR[7]. Despite the critical role of DGCs and/or PDEs in c-di-GMP biosynthesis and/or degradation, mutation or over-expression of some crucial DGCs or PDEs in *Caulobacter crescentus* and *E. coli* hardly affect the global c-di-GMP levels and biofilm formation[8,9]. These observations have led to the hypothesis being put forward that bacteria regulate their behaviors through flexible and specific c-di-GMP signaling processes. More specifically, direct

[1]College of Life Sciences, Northwest University, Xi'an, ShaanXi, China. [2]College of Medicine, Southern University of Science and Technology, Shenzhen, China. [3]Department of Pulmonary and Critical Care Medicine, The First Affiliated Hospital of Guangxi Medical University, Nanning, China. [4]University Laboratory of Metabolism and Health of Guangdong, Southern University of Science and Technology, Shenzhen, China. [5]These authors contributed equally: Xueliang Zhan, Kuo Zhang, Chenchen Wang. ✉e-mail: lianghh@sustech.edu.cn

interactions between DGCs, PDEs, and c-di-GMP receptors play a critical role in this signaling specificity[10]. For example, the local PdeR-DgcM-MlrA[11] signaling module controls the expression of the biofilm regulator CsgD in *E. coli*[12]. DgcC and PdeK mutual interaction controls bacterial cellulose synthase (Bcs)[13–15]. Notably, the interaction hubs involving a few DGCs and PDEs in *E. coli* NosP-NahK and *Pseudomonas fluorescens* mediate c-di-GMP responses[8,16]. Recently, direct interactions among the bacterial DGCs and PDEs have attracted much attention from researchers because these enzymes may perform similar catalytic functions to realize their different cellular decisions.

The enzymatic activity of DGCs and PDEs is activated or repressed under different environmental cues[17]. These enzymes often contain various N-terminal signaling domains such as the HAMP domain, PAS/PAC domain, GAF domain, CACHE domain, and CHASE domain[18]. Each N-terminal signaling domain is able to sense distinct extracellular environmental signals, while the regulatory domain is responsible for the production of signaling molecules, resulting in the altered c-di-GMP levels[19]. For instance, the multidomain transmembrane PA0575 protein in *P. aeruginosa* is an L-arginine sensor that can hydrolyse c-di-GMP[20]. In response to citrate, diguanylate cyclase GcbC, which senses citrate, regulates biofilm formation via enhanced interaction between GcbC and its receptor LapD in *P. fluorescens* Pf0-1[21]. DcpG, a DGC protein, contains a regulatory sensor globin domain linked to GGDEF and EAL domains that are regulated by heme[22]. Though signaling domains in DGCs and PDEs proteins are widely studied, the environmental signals that control c-di-GMP levels and the underlying mechanisms still need to be explored.

Here, we report that the putative CHASE4 domain of IsmP binds to an environmentally relevant molecule, specifically iron. Our results showed that iron/heme inhibited the physical interaction of IsmP and the DGC ImcA and that this inhibited interaction promotes increased c-di-GMP synthesis by ImcA, thereby promoting biofilm formation. In addition, we also described the crystal structures of CHASE4[IsmP] domain and the membrane protein, ImcA, which provides insight into how IsmP stimulates ImcA activity via sensing certain external stimuli. Based on our structural models and corresponding biophysical and biochemical analysis, we proposed a mechanism whereby the crosstalk of these c-di-GMP-metabolizing enzymes is influenced by environmental signals and can adjust their metabolisms to adapt to environmental changes.

## Result

### IsmP interacts with ImcA and other related c-di-GMP metabolic enzymes

An increase in c-di-GMP levels is a major factor in the progression of the *P. aeruginosa* acute infection to chronic[23]. More than 40 genes of *P. aeruginosa* are predicted to be involved in c-di-GMP synthesis or degradation[24]. During the screening of PA2072-specific phenotypes, we surprisingly found that deletion and overexpression of PA2072 gene in *P. aeruginosa* led to biofilm reduction after 14 h of cultivation (Supplementary Fig. 1), a result that is markedly different from those observed with other DGCs and PDEs. Likewise, no wrinkled colonies were observed on the Congo Red (CR) plates (Supplementary Fig. 1). The wrinkled colony phenotype has been attributed to the increased intracellular concentrations of c-di-GMP[25]. Therefore, we hypothesized that these phenotypes, caused by PA2072, are probably due to its interaction with other proteins that are involved in c-di-GMP metabolism, such as the flexible mutual regulation of DgcE-PdeR-DgcM-MlrA complex in *Escherichia coli*[26]. To this end, we performed the bacterial two-hybrid (BTH) assays to determine the direct interaction between PA2072 and 43 other candidate proteins associated with c-di-GMP metabolism (Supplementary Fig. 2a). Notably, PA2072 could interact with 11 proteins and its binding ability with PA1851 was the strongest (Supplementary Table 3). This interaction between PA2072

and PA1851 was further confirmed by the pull-down assay (Fig. 1a), protein co-purification and MicroScale Thermophoresis (MST) (Supplementary Fig. 2b, d). Thus, we devoted our efforts to investigate the biological functions caused by their interaction in the subsequent work. Based on our in-depth analyses of c-di-GMP synthesis, we recommended names for both PA2072 and PA1851 proteins as IsmP (iron-sensing membrane protein) and ImcA (iron modulator of c-di-GMP A), respectively.

Next, the structures of ImcA and IsmP were predicted by SMART (http://smart.embl-heidelberg.de/) (Fig. 1b). Our prediction results reflected that ImcA contains six membrane-spanning domains and a GGDEF domain while IsmP is a multi-domain protein comprised of several domains including the PASPAC (IsmP[273-413]), GGDEF (IsmP[413-585]), EAL (IsmP[585-865]), and CHASE4 (IsmP[59-220]) domains. To further ascertain the nature of their interactions, different truncated domains of IsmP were individually cloned into PUT18C vector (Fig. 1c), and BTH assays were performed. The β-galactosidase activity analyses showed that ImcA and 10 other proteins only interacted with CHASE4_PASPAC (IsmP[1-413]) (Fig. 1c and Supplementary Fig. 2c). Interestingly, PA0847 also contains a CHASE4_PASPAC domain; however, it could not interact with any of the 10 proteins directly bound by IsmP (Supplementary Fig. 2c), suggesting that the CHASE4_PASPAC domain of IsmP along with other 10 proteins exhibiting specific functions in the c-di-GMP regulatory network. Furthermore, BTH screening results revealed that transmembrane regions of ImcA and IsmP are necessary for establishing their mutual interaction (Fig. 1c).

### The DGC activity of ImcA is inhibited by IsmP

ImcA contains a GGDEF domain, which is expected to possess DGC activity for the production of c-di-GMP. However, overexpression of *imcA* in *P. aeruginosa* reduced biofilm and no significant change in bacterial wrinkling was found on CR plates (Supplementary Fig. 3a), indicating that the DGC activity of ImcA was blocked in vivo. As described before, we proposed that IsmP may be required for ImcA's activity by establishing their mutual interaction. Interestingly, we found that overexpression of *imcA* in the Δ*ismP* genetic background (Δ*ismP*/p-*imcA*) significantly promoted biofilm formation and reduced bacterial motility as well as formed wrinkled colonies compared to wild-type cells containing p-*imcA* vector (PAO1/p-*imcA*) after 20 h of cultivation. (Supplementary Fig. 3a, b). A COMSTAT analysis revealed that the Δ*ismP*/p-*imcA* strain significantly increased biofilm biomass and micro-colony height compared to the PAO1/p-*imcA* strain (Fig. 1d). It is well known that biofilm formation and wrinkly colony morphology are associated with c-di-GMP production[27]. To investigate whether the intracellular c-di-GMP accounted for these phenotypes, c-di-GMP levels were monitored in these strains utilizing the p*cdrA-lux* reporter fusion, which is responsive to intracellular levels of c-di-GMP in *P. aeruginosa*[28]. Expression of *imcA* gene in Δ*ismP* mutant (Δ*ismP*/p-*imcA*) exhibited elevated levels of c-di-GMP compared to the wild-type (PAO1/p-*imcA*) (Supplementary Fig. 3c). These results were further validated with the LC-MS/MS experiment, which demonstrated nearly a nearly fivefold increase in the accumulated c-di-GMP concentrations in Δ*ismP*/p-*imcA* strain relative to PAO1/p-*imcA* strain (Fig. 1e). Collectively, these results indicated that the DGC activity of ImcA is inhibited by IsmP in vivo.

As aforementioned, the function of ImcA relies on IsmP in vivo. ImcA contains a GGDEF domain that could convert GTP to c-di-GMP. To further investigate whether IsmP acts as a modulator to regulate DGC activity of ImcA, we purified ImcA, IsmP, and IsmP[CHASE4_PASPAC] proteins (Supplementary Fig. 4c), followed by in vitro enzymatic synthesis of c-di-GMP from GTP utilizing either ImcA alone or both ImcA and IsmP. After incubation, the production of c-di-GMP was evaluated by high-performance liquid chromatographic (HPLC) analysis with retention times consistent with those of GTP and c-di-GMP

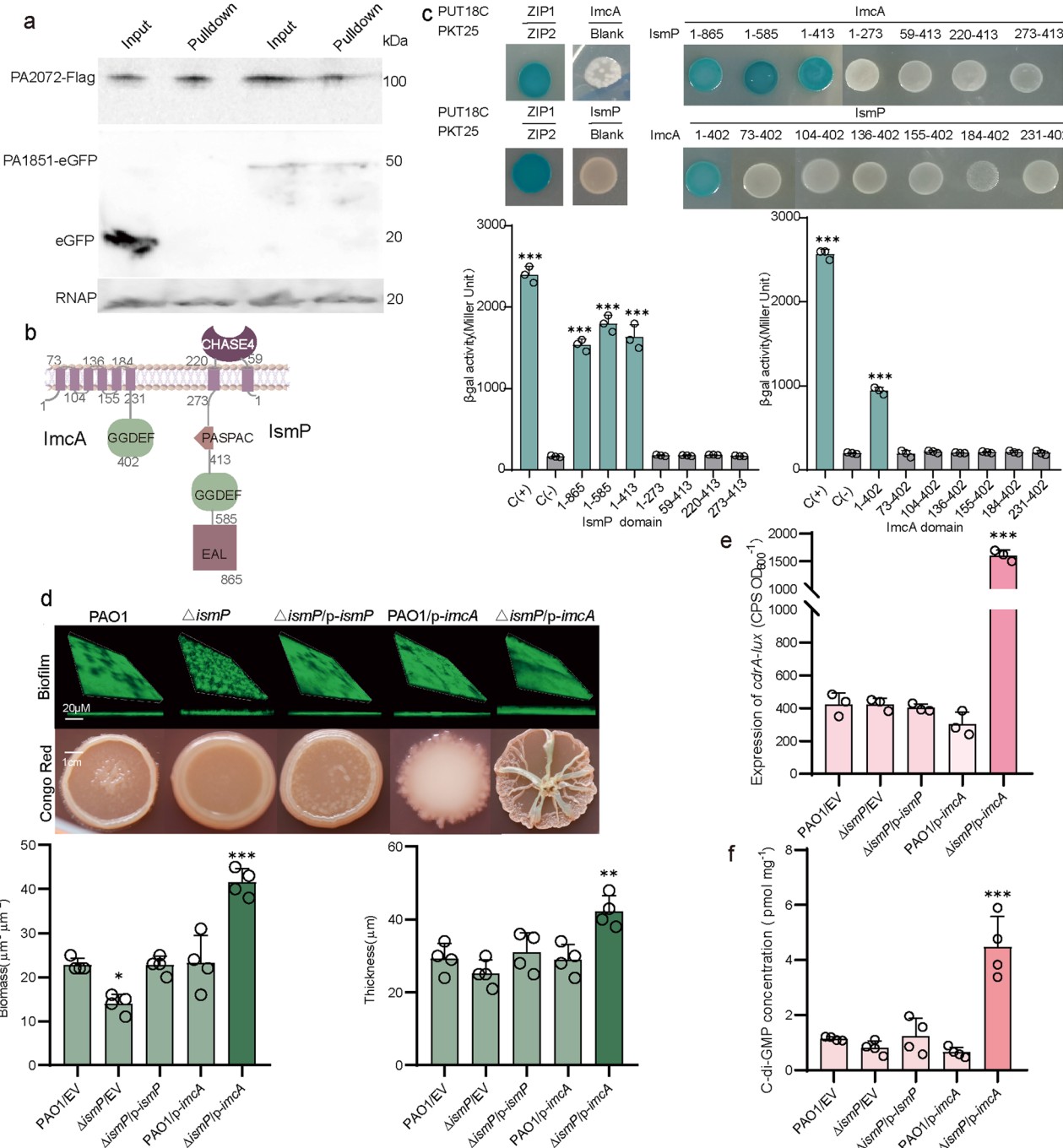

**Fig. 1 | The IsmP-ImcA interaction led to IsmP inhibiting the DGC activity of ImcA. a** Western blot shows IsmP-ImcA interaction. IsmP-Flag co-expressed with ImcA-eGFP or eGFP in PAO1. Supernatant of bacterial lysate incubated with flag beads as input, eluent as pulldown group. This experiment was independently repeated three times with similar results. RNAP was used as a loading control. **b** The schematic representation of the architecture of IsmP and ImcA in PAO1. **c** BTH assays show interactions between full-length IsmP and truncated ImcA fragments, and full-length ImcA and truncated IsmP fragments. The corresponding residue range of truncated IsmP or ImcA indicated above (upper). The quantitative analyses were measured by β-galactosidase activity level (in Miller units) in *E. coli* BTH101 cells (bottom). The interaction between ZIP1 and ZIP2 as a positive control was indicated with C (+), and the negative control was represented by C (-). **d** Biofilm formation (top) and colony morphology (mid) of indicated strains

displayed. Confocal laser scanning microscopy (CLSM) of 36-h biofilms grown in flow cell chambers. The quantitative analyses of biomass and thickness (bottom), which were measured in μm³ μm⁻² and μm, respectively. Colonies morphology of the indicated strains after 2 days of incubation and growth on Congo red plates. **e** Expression of *cdrA-lux* measured in indicated strains after 6 h of cultivation at 37 °C. **f** Measurement of c-di-GMP by LC-MS/MS showed that the Δ*ismP*/p-*imcA* strain produced more intracellular c-di-GMP than the Δ*ismP*/EV strain. Error bars indicate the mean ± s.d. from three or four biologically independent samples. Statistical significance was determined using a two-tailed One-way Analysis of Variance (ANOVA) with a Tukey's multiple-comparison test. In **c**–**e** and **f**, *n* = 3 and *n* = 4 biologically independent samples. *P* values are denoted as * for *P* < 0.05, ** for *P* < 0.01, and *** for *P* < 0.001.

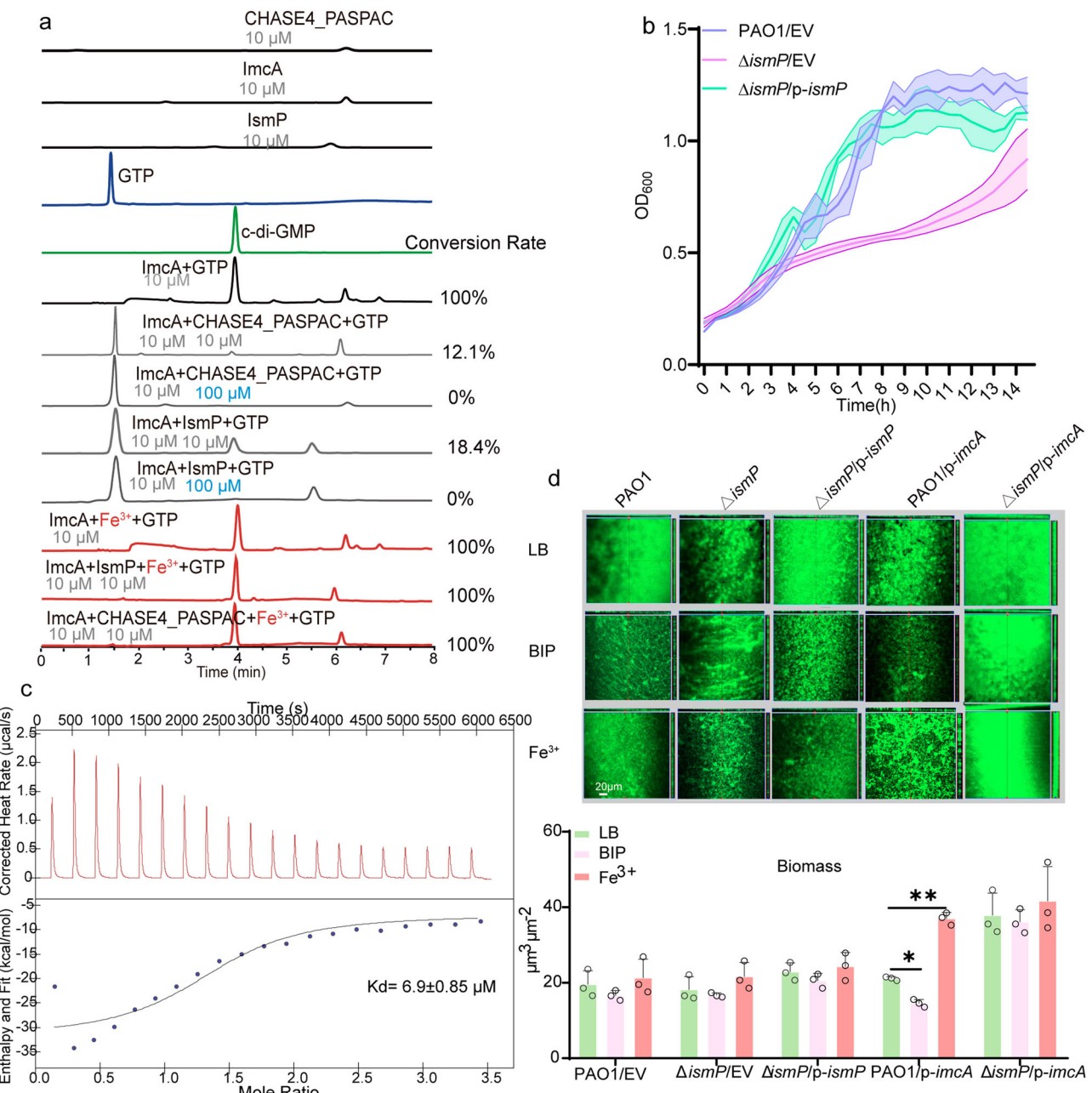

**Fig. 2 | Binding of iron by CHASE4 relieves the inhibitory effect of IsmP on ImcA's DGC activity. a** The catalytic activity of ImcA under either iron-rich or iron-deficient conditions was determined by HPLC. The conversion rates of each reaction have been shown. The concentration of each protein was provided; GTP: 200 μM; c-di-GMP: 100 μM. CHASE4-PASPAC: IsmP[1-413]. **b** Growth of WT, Δ*ismP*, and the complemented strain (Δ*ismP*/p-*ismP*) in the presence or absence of iron (eeM). Purple represents the control group, while red and blue represent the experimental groups. **c** ITC experiment showing CHASE4 (IsmP[40-255]) binding to $Fe^{3+}$. The upper panel represents heat differences upon injection of iron, and the bottom panel shows integrated heats of injection with the best fit (solid line) to a single

binding model using Microcal ORIGIN. **a–c** Data are representative of three independent replicates. **d** CLSM of 36-h biofilms grown in LB medium under the indicated conditions. Green represents the control group, while all other colors represent the experimental groups. Scale bars represented 20 μm. Biomass was measured in $\mu m^3\ \mu m^{-2}$. BIP (2,2'-bipyridyl): an iron chelator. Error bars indicate the means ± s.d of three independent experiments. Statistical significance was determined using a two-tailed One-way Analysis of Variance (ANOVA) with a Tukey's multiple-comparison test. P values are denoted as * for $P < 0.05$, ** for $P < 0.01$, and *** for $P < 0.001$.

standards (Fig. 2a). Remarkably, GTP was absolutely converted to c-di-GMP when GTP was incubated with ImcA alone, indicating that ImcA exhibited DGC activity. However, GTP remained the primary compound in the reaction mix when GTP was incubated with both ImcA and IsmP, which was consistent with the results in vivo (Fig. 2a). Importantly, we found that, similar to that of IsmP, CHASE4_PASPAC also strongly inhibited the DGC activity of ImcA, which further supports the interaction between ImcA and CHASE4_PASPAC domain.

Together, these data suggest that the direct ImcA-IsmP mutual interaction is essential for inhibition of ImcA's DGC activity.

## Binding of iron by CHASE4 domain relieves the inhibitory effect of IsmP on ImcA's DGC activity

Next, we sought to investigate which environmental signals affect the interaction between ImcA and IsmP, thus controlling the intracellular levels of c-di-GMP. Since, CHASE4 domain is predicted to sense

extracellular signals[29], we screened a series of signaling molecules sensed by CHASE4, followed by comparative growth analysis of wild-type and Δ*ismP* strains under various conditions. Interestingly, we found that the Δ*ismP* mutant exhibited a slower growth rate than wild-type PAO1 when these two strains were cultured in BIP medium lacking iron (Fig. 2b), while their growth rate was comparable when cultured in LB medium (Supplementary Fig. 4a). Given that CHASE4 can sense environmental molecules, we next attempted to determine whether CHASE4 bound to iron. The isothermal titration calorimetric (ITC) analysis showed that CHASE4 efficiently bound to trivalent $Fe^{3+}$ with an approximate Kd of 7 μM but it did not bind to $Mn^{2+}$, $Mg^{2+}$ and $Fe^{2+}$ (Fig. 2c and Supplementary Fig. 4b). In *P. aeruginosa* genome, another domain homologous to CHASE4 exists in the PA0847 protein, and it has a weaker binding affinity with an approximate Kd of 16 μM against the trivalent iron (Supplementary Fig. 4e). The results from SEC and native-PAGE experiments show that $Fe^{3+}$ may lead to the aggregation of CHASE4 protein (Supplementary Fig. 4f). In summary, these results clearly suggest that CHASE4 domain is the receptor of iron.

Since IsmP responds to iron through its CHASE4 domain, we hypothesized that iron influences the ImcA-IsmP interaction, subsequently restoring ImcA's ability to synthesize c-di-GMP from GTP. To this end, we measured the biofilm production of the PAO1/p-*imcA* strain under different iron concentrations. Our data showed that the biofilm levels of the PAO1/p-*imcA* strain were decreased in iron-deficient medium while increased in iron-rich conditions (Fig. 2d and Supplementary Fig. 4d). However, the biofilm production of Δ*ismP*/p-*imcA* strain was not affected by the addition of iron (Fig. 2d and Supplementary Fig. 4d). These data indicated that iron could regulate the catalytic activity of ImcA via preventing the ImcA-IsmP complex formation. Furthermore, we evaluated the effect of iron on ImcA and IsmP in vitro. HPLC results demonstrated that ImcA possesses DGC activity and is capable of synthesizing c-di-GMP from GTP (Fig. 2a). We also showed that either ImcA-CHASE4_PASPAC or ImcA_IsmP complex inhibited the catalytic ability of ImcA and thus leading to residual GTP (Fig. 2d). As expected, iron did not affect the catalytic activity of ImcA. However, no GTP was left when iron was introduced into the protein complex (Fig. 2d), suggesting that iron could alleviate the inhibitory effect of IsmP on ImcA's DGC activity.

## Overall structure of CHASE4 and its binding with iron

As previously mentioned, CHASE4 could sense iron to abolish the inhibitory effect of IsmP against ImcA's DGC activity. To further gain insight into how IsmP senses the environment factor-$Fe^{3+}$, extensive crystallization trials were carried out. We solved the crystal structure of CHASE4$^{IsmP}$ at 1.9 Å resolution with $P22_12_1$ symmetry (Supplementary Table 1). As shown in Fig. 3a, each CHASE4 domain contains two discontinuous subdomains: the membrane-distal subdomain and the membrane-proximal subdomain connected by a loop. The former exhibits an alpha+beta fold pattern, which contains a core five-stranded β-sheet, with other helices distributed on both sides of β-sheet; the latter mainly contains a three-stranded β-sheet, which is connected with the former through short α8 helix (Fig. 3a). As shown in Fig. 3b, the dimeric form is maintained by a pseudo-symmetric interface mainly consisting of several polar bonds mediated by Q51, W55, K96, 97 N, T180, D182, W212 and P247 from either chain in a dimer (Fig. 3c and e). In addition, the sidechain of N107 interacted with the Q69 from the other monomer via a hydrogen bond (Fig. 3d). Furthermore, the native-PAGE and SEC results indicated that CHASE4 domain exists as a dimeric form in solvent. However, it could further oligomerize when expressed in $Fe^{3+}$-containing M9 medium (Supplementary Fig. 4f). ICP-MS analysis of the element highlighted the existence of $Fe^{3+}$ in CHASE4 sample (Fig. 3f). We have tried several times to obtain the crystal structure of CHASE4-$Fe^{3+}$ complex but it failed. Due to advances of computational biology, a putative $Fe^{3+}$-bound CHASE4 rough model could be established based on in silico analysis (Fig. 3g).

Clearly, at least three amino acids, including E65, N66, and Q69 may play a key role in the $Fe^{3+}$ binding to CHASE4 according to this model. To verify this hypothesis, a series of single and multi-site directed mutants were constructed. ITC assays demonstrated stronger binding affinity of IsmP$^{CHASE4}$ towards $Fe^{3+}$, however, the IsmP$^{CHASE4}$ mutants exhibited weaker, even no binding affinity with $Fe^{3+}$, suggesting the critical role of E65, N66, Q69 in $Fe^{3+}$ sensing and ultimately binding (Fig. 3h and Supplementary Fig. 4g). In addition, we performed biofilm formation assay to confirm whether binding of iron by CHASE4 mutants affected the inhibitory role of IsmP on ImcA's DGC activity when alanine substitution occurred in the potential $Fe^{3+}$ binding pocket. Unlike wild-type *ismP*, the biofilm production was not induced by iron in Δ*ismP*/p-*imcA* strain overexpressed with *ismP*$^{E65A/N66A/Q69A}$ (Fig. 3i). Together, here we reported the crystal structure of CHASE4$^{IsmP}$ domain with no homolog deposited in PDB, and further confirmed its potential $Fe^{3+}$ binding pocket utilizing ICP-MS, ITC and biofilm formation assays.

## ImcA comprises a stable six-transmembrane domain and a relatively liable DGC catalytic domain

To further investigate the interaction mode between ImcA and its substrate, the substrate analog GMPCPP-bound ImcA homodimer structure was determined using Cryo-EM at 3.07 Å resolution (Fig. 4a and Supplementary Fig. 5a-f). ImcA is primarily composed of two parts, such as the N-terminal 6TM domain spanning from L47 to E231 and the C-terminal GGDEF (equivalent to DGC) domain containing residues from R241 to A393, which are connected to each other by a flexible hinge loop (Fig. 4a and Supplementary Fig. 6a). The latter folded into a typical GGDEF pattern consisting of a five-stranded antiparallel β-sheet (β1-β5) surrounded by five discontinuous α helices, which is basically conserved throughout the DGC proteins (Supplementary Fig. 6b-c).

Our PISA analysis depicted that the inter-ImcA monomer interface covered an area of ~2656.4 Å$^2$ with a solvation-free energy gain ($\Delta^i G$) of -48.6 kcal/mol, which corresponds to a strongly hydrophobic interface. Apparently, a pseudo-dyad axis could be found in the interface and hydrophobic van der Waals interactions contributed a lot to the interface (Fig. 4a-b). In addition, E218 located in chain A interacted with Q219 and K223 from chain B by forming polar bonds, which is entirely equivalent to the way how E218 from chain B interacted with Q219 and K223 situated in chain A (Fig. 4b). Another symmetric contacts could be observed on the extracellular side of 6TM domain, namely, the sidechain of T190 in chain A formed a hydrogen bond with the backbone of F141 situated in ECL2 of chain B, so it's the same case with F141 in chain A and T190 from chain B (Fig. 4b). It could be inferred that all contacts mentioned above are actually involved in maintaining the stability of ImcA homodimer. Nevertheless, it's worth noting that no polar contact between C-terminal DGC domains was observed (Fig. 4c). And considering the existence of long flexible hinge loop, ImcA DGC domains presumably may keep undergoing high-speed vibration in a local space as reflected by its higher overall B factor and worse electron density map compared to 6TM domain (Supplementary Fig. 5c).

## Structural basis for ImcA-catalyzed GTP condensation to synthesize c-di-GMP

Given the symmetry, only one single substrate binding pocket will be thereby described in detail here (Fig. 4c-d and Supplementary Fig. 8a-e). The hydrophobic pocket was predominantly formed by α10-α11 and α14 helices together with β1-β3 strands of five-stranded antiparallel β-sheet (Fig. 4d and Supplementary Fig. 7a). As depicted in Fig. 4d, the nonhydrolyzable analog "GMPCPP" of substrate GTP adopted a relatively extended conformation when bound to the catalytic pocket. Despite the hydrophobicity, several polar residues were found in the pocket and its vicinity. For instance, the nitrogen atom from the sidechain amino group of R241 in chain A formed hydrogen bonds with

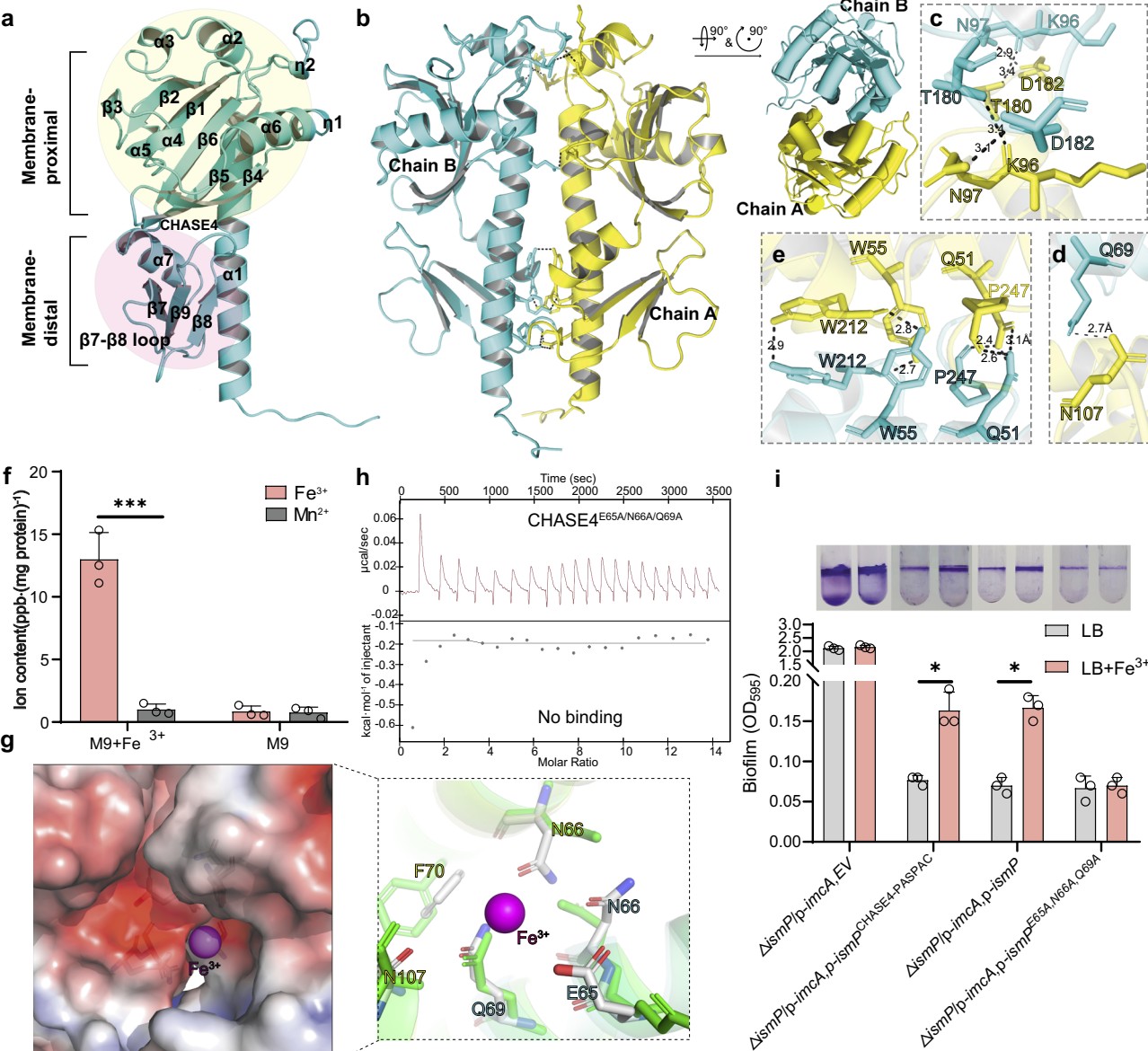

**Fig. 3 | The crystal structure of CHASE4 and the model of its binding with iron.**
**a** The IsmP CHASE4 domain encompasses a membrane-proximal subdomain and a membrane-distal one. The secondary structure units are defined by DSSP algorithm and labelled with black. **b** Two views of CHASE4 homodimer, which is maintained by a symmetric interface that consists of three inconsecutive contacts (**c**–**e**). The residue labels are allocated successively with the single-letter identifier of certain amino acid and its corresponding residue number, and colored in corresponding color. And the polar bonds are indicated with dashed lines with corresponding distances labelled in black (similarly hereinafter). **f** ICP-MS showing the purified CHASE4 protein containing iron. **g** The predicted $Fe^{3+}$ (magenta sphere) binding pocket of CHASE4 with the electrostatic potential surface shown (left) and whose enlarged details (right). Even though the $Fe^{3+}$-bound model was refined by

HADDOCK 2.4, the side chains located in the predicted pocket may still not have the correct geometry or bonding distances with $Fe^{3+}$. The sidechains of residues located in the pocket are represented by stick with the atom colored by element (N: blue, O: red, C: white) and the crystal apo structure is shown with green cartoon (backbone) or sticks (side chains). The surface was generated by APBS (Adaptive Poisson-Boltzmann Solver), whose level ranges from −5 to 5, and the corresponding surface is colored from red to blue gradually. **h** ITC showing the CHASE4$^{E65A/N66A/Q69A}$ mutant could not bind to iron. **i** Biofilm formation of the indicated strains grown in LB supplemented with or without iron was displayed with crystal violet staining (up) and quantified with optical density measurement (down). In **f** and **i**: Error bars indicate the means ± s.d. ($n = 3$); *$P < 0.05$, ***$P < 0.001$ based on one-way ANOVA test.

the N3 atom of purine ring and the 2'-hydroxy group (2'-OH) oxygen atom of ribofuranose, respectively (Fig. 4d). And the 6'-hydroxy group (6'-OH) oxygen atom in purine ring could interact with the sidechain amino group of R308 in chain B via a polar bond (Fig. 4d). The sidechain of D285 from α12 helix of chain B interplayed with the 2'-amino group (2'-NH2) of guanine ring via a hydrogen bond (Fig. 4d). Moreover, the ribofuranose 3'-hydroxy group (3'-OH) also formed a hydrogen bond with the sidechain carbonyl oxygen atom of N276 in chain B. Consequently, GMPCPP was finally attracted and anchored to a narrow stripped positively-charged surface with its purine ring

stretching to the hinge loop, meanwhile, crossing the top of the conserved $^{310'}$GGEEF$^{314'}$ motif in DGC domain (Fig. 4d). Whilst the sidechain nitrogen atoms of K273 and K383 together with the carboxyl oxygen atom of E312 located in chain B established polar bonds with oxygen atoms from α、β or γ phosphate groups, respectively. The ligand was accordingly further stabilized, of which negative charges carried by phosphate groups were partially neutralized by K273 and K383 (Fig. 4d). Residue mutation analysis in PyMOL revealed that any residue substitution of G$^{311}$ located in $^{310'}$GGEEF$^{314'}$ motif will compress the space where substrates will take up. In contrast, G$^{310}$ substitution

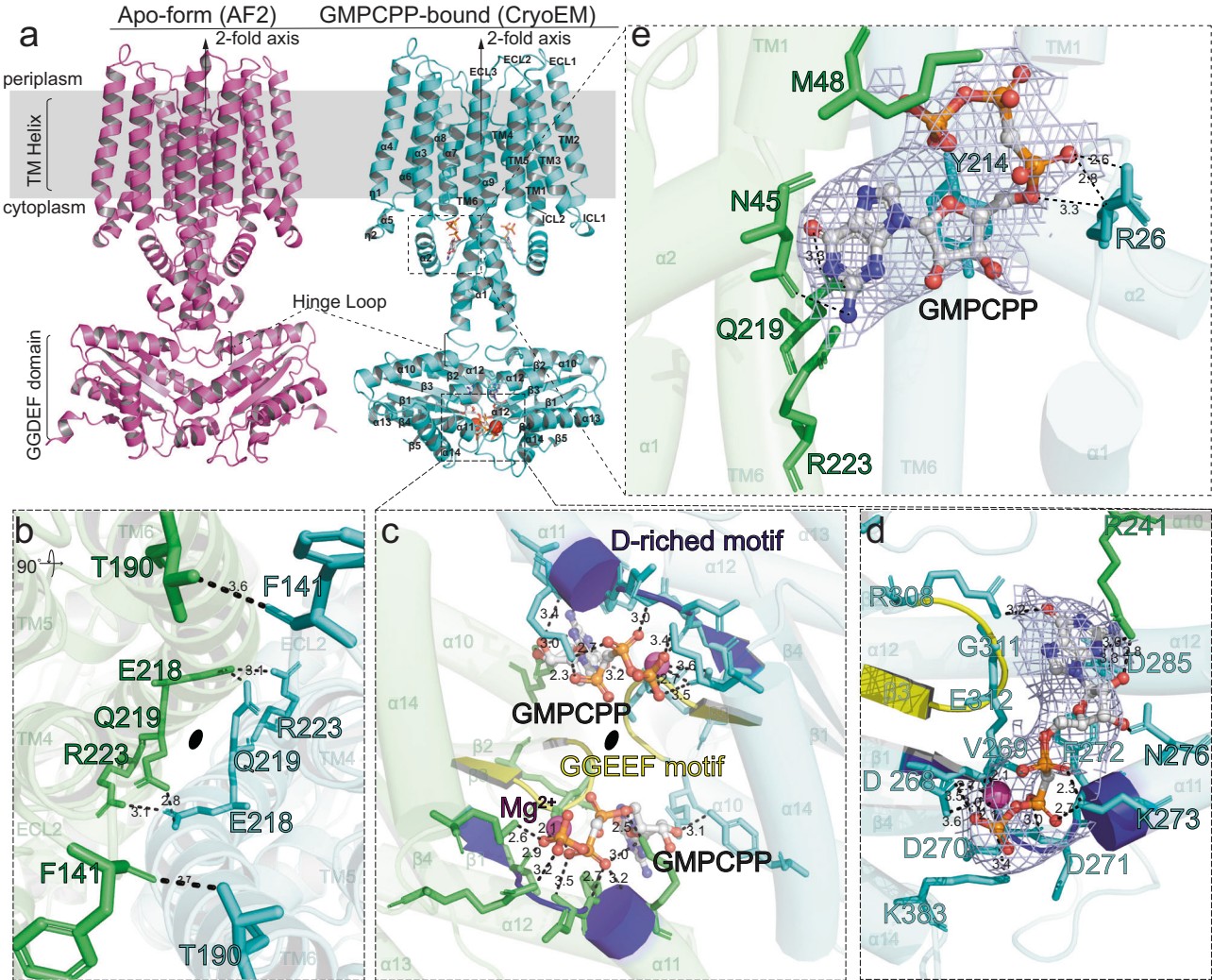

**Fig. 4 | ImcA is DGC protein with a six-TM domain. a** ImcA homo-dimeric apo form predicted by AlphaFold2 (left) and its substrate analog GMPCPP-bound form (Cryo-EM, right). The secondary structure units are defined by DSSP-Stride plugin in PyMOL. And the ligand was represented with ball-and-stick, and further colored by element (N: blue, O: red, C: white, P: orange). **b** There is a symmetric contact interface between TM domains. The 2-fold rotation axis is indicated by a black solid ellipse and the same below. The residue labels are composed successively of the single-letter identifier of certain amino acid and corresponding residue number, and colored in the same color with corresponding chain. **c** Details of the dimeric catalytic domains with the secondary structure units of each chain colored in corresponding color. The sidechains of amino acids situated in the vicinity of catalytic pocket are displayed with stick. Moreover, the GGEEF motif and identified D-riched motif are colored in yellow and blue, respectively. **d** Enlarged details of single ImcA DGC catalytic domain located in GMPCPP-bound dimeric form with the electron density map for GMPCPP (8.0σ) shown in light blue meshes. **e** Enlarged details of the membrane-proximal ligand binding pocket with the electron density map for GMPCPP (6.5σ) shown in light blue meshes.

tends to make DGC domains far from each other, which will also inhibit DGC activity. Furthermore, $Mg^{2+}$ was not stabilized by β and γ phosphate groups of GMPCPP, but also $D^{268}$ and $V^{269}$ in D-riched motif $^{268'}DVDDF^{272'}$ together with $E^{312}$ from $^{310'}GGEEF^{314'}$ (Fig. 4d). It's rational to infer that $^{310'}GGEEF^{314'}$ and $^{268'}DVDDF^{272'}$ together with $Mg^{2+}$ presumably played a pivotal role in catalyzing the formation of cGMP moieties and their cyclization to produce second messenger C-di-GMP when bound to a hydrolysable substrate.

Interestingly, there are two extra symmetric ligands found in the vicinity of the cytoplasmic side of 6TM domain, which is not reported earlier (Fig. 4e and Supplementary Fig. 8f-j). Compared to those located in the DGC catalytic pockets, the GMPCPP adopted a relatively bent conformation, interacting with fewer residues via polar bonds mediated by R26 and N45 located in front of TM1, Q219 and Y214 located on TM6 (Fig. 4e and Supplementary Fig. 7b). The reduction of biofilm production by directed alanine substitution mutations further verified the existence of ligand here (Supplementary Fig. 7c). These results revealed that mutants could lead to the decreased cyclase catalytic activity, further inhibiting the synthesis of c-di-GMP, which is consistent with the BTH assays (Supplementary Fig. 7c and Fig. 1c). Taken together, we determined the cryo-EM structure of ImcA containing 6TM domain and revealed the importance of TM domain for its DGC activity.

## A proposed ImcA-IsmP₂-ImcA hetero-tetramer complex structural candidate model

The defect of EAL and the degenerate GGDEF domain of IsmP make little impact on its interaction with ImcA$^{FL}$. In contrast, a defect in the PAS domain abolished IsmP-ImcA mutual interaction (Fig. 1c), suggesting the importance of PAS$^{IsmP}$ to DGC$^{ImcA}$ activity inhibition. Similarly, the deletion of any TM helix from IcmA also disrupted the interaction and subsequently lose DGC activity (Fig.1c and Fig. 5a). Due to the presence of detergent DDM, whose single micelle molecule is about 70 kDa$^{30}$, it seems extremely difficult to calculate the exact relative molecular weight (Mr) of ImcA-IsmP complex. However, it could be inferred that the ImcA-IsmP polymer might be a hetero-

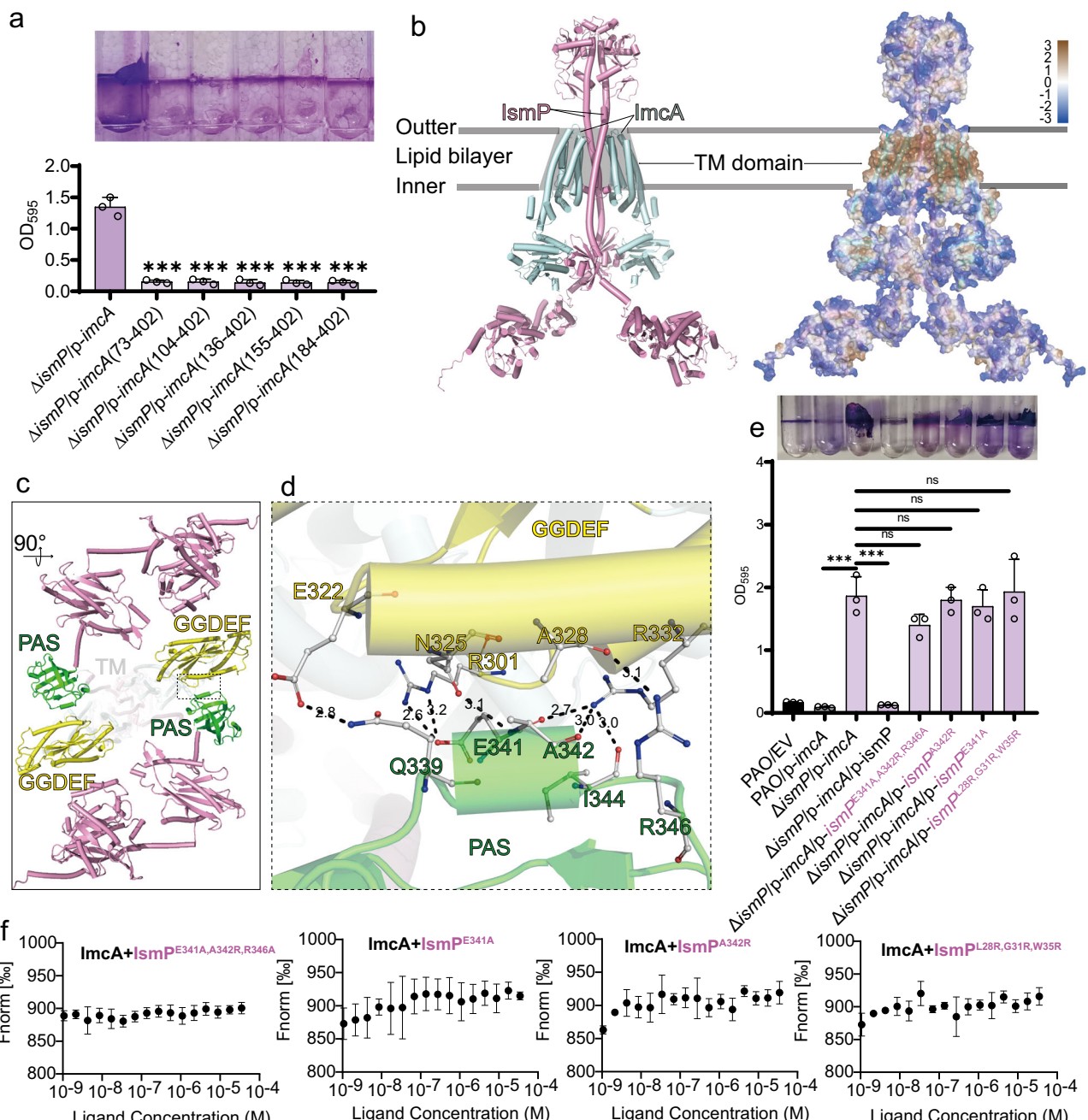

**Fig. 5 | IsmP inhibits ImcA's DGC activity by its PAS domain interacting with GGDEF domain of ImcA. a** The transmembrane domain of ImcA is necessary for its DGC activity. Biofilm formation of the Δ*ismP* strain expressing various truncated *imcA* was displayed with crystal violet staining (up) and quantified with optical density measurement (down). **b** The docking model of ImcA-IsmP₂-ImcA hetero-tetramer complex (left) with corresponding hydrophobicity surface detected using Kyte-Doolittle hydrophobicity scale (probe radius=1.4 Å) displayed on its right side. **c,** Another view of of ImcA-IsmP₂-ImcA heterotetramer complex in panel **b**. The PAS^IsmP domains and GGDEF^ImcA domains are highlighted in green and yellow, respectively. One of the symmetric contact interfaces formed by PAS^IsmP and

GGDEF^ImcA is besieged with rectangular black dashed lines and further enlarged in **d** panel. **d** The residue sidechains situated in the heterodimer interface are shown as ball-and-stick and colored by element. The polar bonds are represented with black dashed lines with their corresponding distance indicated in black. **e** The designed single and multi-site directed mutation assays and their corresponding post-mutational biofilm phenotypes are displayed above. **f** MST analysis shows no binding of ImcA with these IsmP mutants, this experiment was independently repeated three times with similar results. In **a** and **e**: Error bars indicate the means ± s.d. of three biological replicates. ns: no significance; (*n* = 3); *** for *P* < 0.001.

tetramer based on the SEC result (Supplementary Fig. 2b). To further acquire insights into the interaction mode of IsmP and ImcA, molecular docking combined with simple molecular dynamics simulation was used to generate a possible structural candidate model of ImcA-IsmP₂-ImcA hetero-tetramer (Fig. 5b). Given the instability of SEC method, the real oligomeric form may not be consistent with the tetramer, however, a ImcA-IsmP hetero-oligomer is present. The

predicted hetero-dimer interface in the model highlighted a strong interaction between ImcA^DGC and IsmP^PAS and the considerable van der Waals interactions between their TM domains (Fig. 5b, c). Specifically, residues E341, A342 and R346 in IsmP^PAS could establish hydrogen bonds and/or salt bridges with R301, N325, A328 from ImcA^GGDEF, respectively (Fig. 5d). In addition, Q339 and I342 in IsmP^PAS also participated in the interface recognition by forming polar bonds with E322

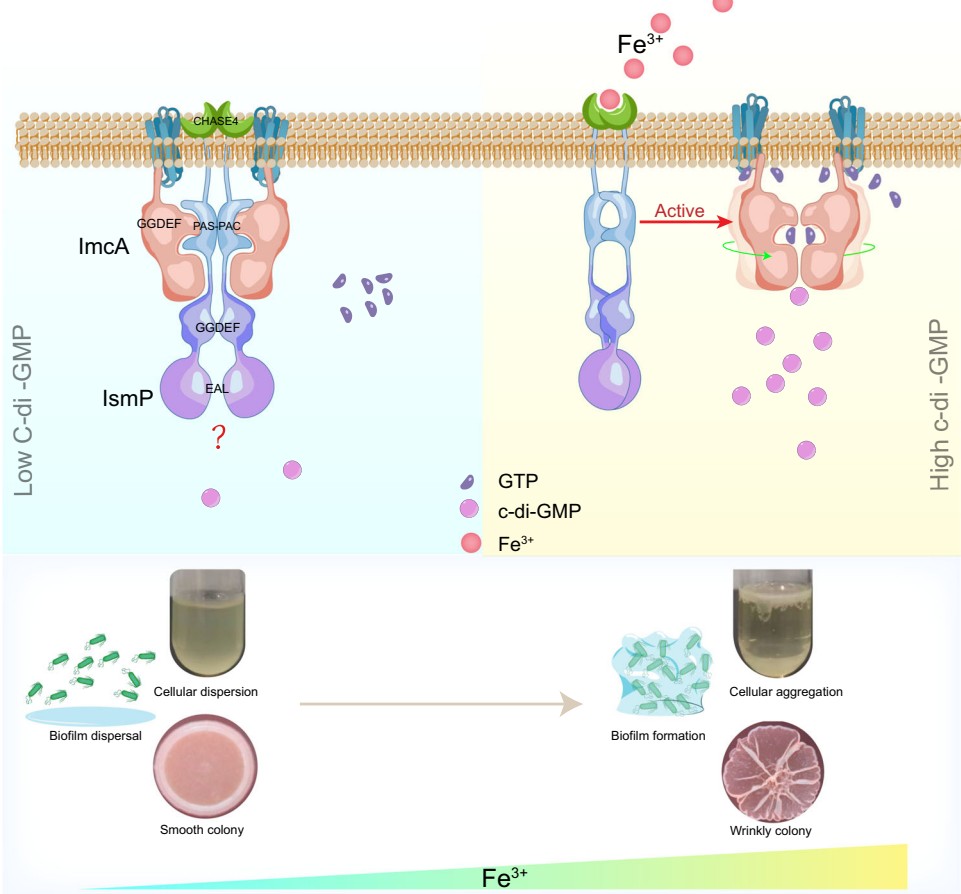

**Fig. 6 | Proposed model of the IsmP-ImcA signaling switch in response to iron involved in regulation of *P. aeruginosa* behaviors.** Under iron-defective conditions, IsmP directly interacts with ImcA, which results in the inhibition of ImcA's DGC activity and produces low intracellular c-di-GMP levels. Conversely, when under iron-rich conditions, CHASE4 senses iron to alleviate the inhibitory effect of IsmP on ImcA and thus stimulates the DGC activity of ImcA. The high concentrations of c-di-GMP tend to form cellular aggregation, wrinkly colony, and biofilm. Notably, GTP is also accumulated near the cell membrane in proximity to ImcA, which participates in the synthesis of c-di-GMP.

and R332 in GGDEF[ImcA] domain, respectively (Fig. 5d). Site-directed mutations (L28R, G31R and W35R) in IsmP TM helix could abolish the interaction between IsmP and ImcA, further liberating the ImcA DGC activity (Fig. 5e, f). More or less the same trend of interaction behavior was also observed for designed E341A, A342R, R346A directed mutation in the predicted interface (Fig. 5e, f). Abovementioned results based on mutants further confirm the key residues and the existence of hetero-dimer interface. And it seems as if the hetero-dimer (PAS[IsmP]-DGC[ImcA]) interface couldn't be formed without the disassociation of ImcA homodimer. Overall, a possible candidate hetero-tetramer model for ImcA-IsmP complex was proposed here, which may help to better understand the structural basis of how IsmP inhibits ImcA DGC activity, especially the ImcA-IsmP hetero-dimer interface which was verified by mutant phenotype and biophysical method (Fig. 5e, f).

## Discussion

Bacterial c-di-GMP is a widespread second messenger responsible for regulating many critical physiological functions such as biofilm formation and virulence[13]. Although c-di-GMP downstream signaling pathways, including identification of c-di-GMP receptors and c-di-GMP-regulated processes have been extensively investigated[5], less is known about the molecular mechanisms for sensing the environmental signals to regulate the cellular c-di-GMP levels in bacteria[31–33]. Previous research demonstrated that signal-sensing structural domains such as GAS, PAS, and CHASE are often found at the N-terminus of c-di-GMP metabolic enzymes (CMEs)[34]. These sensory domains can directly respond to various signal stimuli, including gases, light, redox state, temperature, and chemical compounds[32]. For instance, the DGC activity of CasA in *Vibrio fischeri* is stimulated in response to calcium[35]. The PAS domain mediates light and redox sensing through flavin co-factors[36]. The CHASE4 domain is found in multiple transmembrane receptors upstream of bacterial signal transduction pathways[37]. In bacteria, CHASE4 is almost exclusively found in diguanosine metabolism-related enzymes[38]; however, the signal molecules recognized by CMEs[CHASE4] domain still remains enigmatic.

Here, we unveil that IsmP[CAHSE4] domain is the receptor of iron. Binding of iron to CHASE4 weakens the interaction between IsmP and ImcA and diminishes the inhibitory effect of IsmP on ImcA, which activates the DGC activity of ImcA, causing an increase in the intracellular c-di-GMP levels, and ultimately leading to biofilm formation (Fig. 6). We found heme, which contains iron, can also produce the same biological phenotype as iron (Supplementary Fig. 4d and Supplementary Fig. 9c), indicating that heme may also be another environmental effector for CHASE4. It has been shown that *P. aeruginosa* acquires trivalent iron and heme from the blood via various carriers[39]. Inside the cell, $Fe^{3+}$ is reduced to $Fe^{2+}$, participating in biochemical reactions and physiological processes[40,41]. Moreover, the trivalent iron concentration in the respiratory tract of cystic fibrosis patients exceeds that of healthy individuals, surpassing 8 μM[42,43]. Interestingly, only PA0847 and IsmP proteins in *P. aeruginosa* PAO1 genome possess highly similar CHASE4 domains. Our ITC data depicted that both IsmP

and PA0847 have the capacity to bind iron with approximate Kd values of 7 and 16 μM, respectively (Fig. 2c and Supplementary Fig. 4e), which emphasizes the importance of CHASE4 domain in sensing distinct environmental stress stimuli. Despite the presence of CHASE4 domain in PA0847, it was unable to establish its mutual interaction with ImcA, suggesting the existence of additional PA0847-mediated signal factors might be required to maintain the iron susceptibility in *P. aeruginosa*. Further research is required to draw complete picture of the molecular mechanism of the environmental stress-mediated signaling pathway. Given the wide distribution of CHASE4 homologs in bacteria (Supplementary Fig. 10) and its binding with iron, it is suggested that may *Proteobacteria phylum* could quickly adapt to the environmental stresses or nutritional starvation via CHASE4 sensing to signaling molecules.

Our previous results suggested that the inhibition of ImcA DGC activity by IsmP is essential to decrease the overall motion ability of DGC domain, which is required by catalytic activity (Fig. 5b–d). To further get insights into the flexibility together with potential structural differences induced by the substrate binding, the dimeric ImcA DGC domain predicted by AlphaFold2 was superposed with the corresponding ligand-bound form (CryoEM) and DgcZ protein (the prototype for DGC protein) from *E. coli* (Supplementary Fig. 11). The results highlighted structural conformational changes in the residue sidechains (red) located in the catalytic pocket, and especially the shift between DGC domains upon substrates binding to catalytic pockets. In detail, the gap between two ligand-bound DGC domains was smaller than those in apo form as well as DgcZ (Supplementary Fig. 11). However, the distance between atoms required for covalent bonds with C-di-GMP was slightly larger than DgcZ, which indicate that the DGC domains probably require much stronger motion ability to produce C-di-GMP, which is consistent with the feature of long hinge loop. Higher overall B factor and worse electron density map also verified the overall flexibility of DGC domain pair (Supplementary Fig. 5c). Moreover, the deletion of TM helices will directly result in the inability of diguanylate cyclase (Fig. 5a). The extra ligand binding sites located in the 6TM domain suggested that the membrane-proximal part of which may participate in the catalytic procedure. Overall, it could be inferred that the substrate-free DGC domains in inactivated dimeric ImcA were far away from each other, which is principally caused by the flexibility of hinge loops and lack of contacts between DGC domains. However, the positively charged cytoplasmic side of 6TM attracted substrates to aggregate around the ImcA. Consequently, substrates could easily slide into DGC catalytic pockets and two substrate-loaded DGC domains could further move into a catalytically competent dimeric arrangement (Fig. 6).

C-di-GMP has been studied for decades and great progress has been made in understanding the nature of c-di-GMP receptors and the regulatory functions of c-di-GMP. However, studies regarding the upstream signals affecting CMEs are relatively limited. CMEs usually sense environmental stimuli in either one-component[44] or two-component systems[45,46]. For instance, PDE activity of DosP in *E. coli* is stimulated via ligand binding to the heme-Fe (II) complex[47] Likewise, our data reflected that iron promotes the DGC activity of ImcA via weakening the interaction between IsmP and ImcA, which is similar to the mechanism of NO-induced biofilm dispersal via the NosP-NahK signal axis in *Legionella pneumophila*[16]. It should be noted that this mechanism and the one involving IsmP and ImcA, which are two distinct CMEs, are not exactly the same. Additionally, our data demonstrated that IsmP and ImcA can efficiently bind 11 and 6 CMEs, respectively. Although all exact input signal factors have not been completely characterized, the complex interactions between CMEs contribute to various modes of c-di-GMP signaling (Supplementary Fig. 9a, b). In fact, many bacteria maintain remarkably low intracellular c-di-GMP concentrations at about 60 to 150 nM in the cell[48,49]. Our LC-MS analysis showed similar c-di-GMP concentration in both PAO1 and

PAO1/p-*imcA* strains, while 5-folds increase was observed in Δ*ismP*/p-*imcA* strain (Fig. 1f). However, this result is not due to an increase in the expression of *imcA* in Δ*ismP* strain (Supplementary Fig. 9d). Although the global cellular c-di-GMP level is relatively lower in the Δ*ismP*/p-*imcA* cells than that in single DGC knockout mutants such as *sadC* and *wspR*[7,50], it can produce many biofilms and significantly inhibit bacterial motility (Supplementary Fig. 3a, b). Since the increasing concentration of c-di-GMP causes more effectors to bind, we reasoned that the increased c-di-GMP in Δ*ismP*/p-*imcA* cells can bind the uncharacterized effectors, ultimately leading to altered bacterial behaviors.

We have found that IsmP and ImcA interact with 10 and 6 c-di-GMP metabolic enzymes, respectively (Supplementary Fig. 9b), which composed a complex c-di-GMP regulatory network. In order to determine the effect of iron and ImcA on biofilm formation via interacting with IsmP, we have tried to find out an appropriate concentration of $Fe^{3+}$ for this experiment. We measured biofilm production of WT PAO1, Δ*ismP*/EV, Δ*ismP*/p-*ismP*, PAO1/p-*imcA* and Δ*ismP*/p-*imcA* in LB medium supplemented with different concentrations of $Fe^{3+}$. It is noted that biofilms of WT PAO1 overexpressing *imcA* are significantly increased but the WT strain is not impacted by changes to 100 μM $Fe^{3+}$ (Supplementary Fig. 4d), which suggests that $Fe^{3+}$ plays a critical role in stimulating the DGC activity of ImcA. Importantly, our data showed that, like PAO1/p-*imcA*, the biofilm production of WT PAO1/EV is also significantly increased in LB medium with addition of 200 μM $Fe^{3+}$. However, no difference was observed for the Δ*ismP*/EV strain (Supplementary Fig. 4d), indicating that iron regulates biofilms via IsmP and dependent on the concentrations. Combined with our biochemical data, these findings pinpoint that binding of $Fe^{3+}$ with IsmP alleviates the IsmP-ImcA interaction and then promotes the DGC activity of ImcA (Fig. 2a), which provides insight into how *P. aeruginosa* flexibly regulates intracellular c-di-GMP to control specific effector/target systems.

Based on our structural characterization combined with the corresponding biochemical experiments, we proposed that *P. aeruginosa* uses the IsmP-ImcA module as a flexible dynamic regulation mechanism to mediate the molecular switching of acute to chronic infection via sensing the extracellular signals. Multiple DGCs or PDEs sometimes produce highly specific regulatory outputs—even in the presence of other DGCs or PDEs simultaneously[19]. Alternatively, *P. aeruginosa* also use c-di-GMP signaling to produce relatively low c-di-GMP levels depending on the actual conditions[13]. The present work provides an example of how *P. aeruginosa* can flexibly regulate intracellular c-di-GMP in controlling particular effector/target systems based on distinct iron concentrations, thus regulating the transition from acute to chronic infections. Given the mechanistic versatility of c-di-GMP signaling and the multiplicity of CMEs involved in many bacterial species, this work will help refine the complex of c-di-GMP regulatory networks and improve our understanding of how bacteria dynamically switch and adapt to changing conditions via signaling domains.

## Methods

### Bacterial strains

The bacterial strains and plasmids used in this study are listed in Supplementary Table 4. The primers used in this study are listed in Supplementary Table 5. Cells were grown in lysogeny broth (LB). LB agar was used as a solid medium. When required, antibiotics were added to these media at the following concentrations: 100 μg mL⁻¹ ampicillin,10 μg mL⁻¹ tetracycline and 25 μg mL⁻¹ streptomycin for *E. coli*; and 300 μg mL⁻¹ carbenicillin, 50 μg mL⁻¹ gentamicin and 100 μg mL⁻¹ tetracycline for *P. aeruginosa*.

### Construction of plasmids

DNA cloning and plasmid preparation were performed according to standard methods. PCR primers were designed with restriction sites

at their ends for subsequent digestion and ligation into the specific vector (Supplementary Table 4 and Supplementary Table 5). For constructing pET-28a-CHASE4, the primer pair CHASE4f and CHASE4r was used to amplify the CHASE4 (residues 59-220 of PA2072) fragment from the genomic DNA. The PCR products were digested and inserted into pET-28a to generate pET-28a-CHASE4. Other plasmids were constructed using a similar method. For construction of the pET-CHASE4$^{E65A/N66A/Q69A}$ vector, the primer pair pET-CHASE4$^{E65A/N66A/Q69A}$f and pET-CHASE4$^{E65A/N66A/Q69A}$r was used to amplify the CHASE4$^{E65A/N66A/Q69A}$ fragment using pET-28a-CHASE4 as template, and then the PCR products were ligated using the Seamless cloning kit. Other site-mutagenesis vectors were generated with the same method. In order to construct the Mini-CTX-ismP-Flag plasmid, the Mini-CTX-ismP-FLAG f and Mini-CTX-ismP-FLAG r primers were used to generate the ismP containing its promoter fragment by PCR. The digested PCR product was then cloned into the Mini-CTX-Flag plasmid. Similarly, to construct the Mini-CTX-ismP-FLAG-imcA-eGFP plasmid, the Mini-CTX-imcA-eGFP f1, Mini-CTX-imcA-eGFP r1, Mini-CTX-imcA-eGFP f2, Mini-CTX-imcA-eGFP r2 primers were used on the DNA fragment encoding the imcA-eGFP gene driven by the imcA promoter, which was generated through overlapping PCR. The digested PCR product was then cloned into the Mini-CTX-ismP-Flag plasmid. Restriction and DNA-modifying enzymes were used following instructions from the manufacturers. The instructions for seamless cloning were followed to construct the vector. Transformation of *E. coli* DH5α, *E. coli* C43 (for cloning) and *P. aeruginosa* was carried out by electroporation. All plasmids were verified by sequencing. All mutations were confirmed by DNA sequencing.

## Construction of *imcA* and *ismP* deletion mutants

In this comprehensive account, we meticulously construct gene knockout mutants employing a sacB-based strategy. The process specifically involves the generation of null mutants for ismP and imcA (ΔismP and ΔimcA). This is achieved by amplifying sequences both upstream and downstream of the targeted deletion, each sequence encompassing approximately 1500 base pairs. This method facilitates precise manipulation of the genetic material, ensuring accuracy in our experimental procedures. This amplification is performed on PAO1 genomic DNA, employing primer pairs KO1851F1/R1, KO1851F2/R2 and KO2072F1/R1, KO2072F2/R2 (Supplementary Table 2). Subsequent to digestion, these PCR products are cloned into the pEX18Ap vector[51], culminating in the formation of pEX18Ap-ismP and pEX18Ap-imcA. The successfully constructed plasmid was electroporated into PAO1 and plated onto a culture dish containing 300 μg mL$^{-1}$ carbenicillin. After 2 days, monoclonal clones were isolated and streaked onto an LB culture dish containing 20% sucrose. The mutants were verified by PCR and sequencing.

## Isothermal titration calorimetry (ITC) assays

Isothermal titration calorimetry experiments were performed at 25 °C with a MicroCal VP-ITC instrument (Malvern Panalytical). Use 50 μM CHASE4 protein and titrate with 250 μM ion solution until saturation is reached. CHASE4 and ion solution were diluted in the same buffer: 20 mM HEPES pH 7.5, 200 mM NaCl. Inject 0.5 μL each time for a total of 20 injections, discarding the first injection when analyzing the data. Data were processed with the MicroCal VP-ITC Analysis software (Malvern Panalytical) and fitted with the 'single set of identical sites' model.

## Bacterial two-hybrid (BTH) detection of protein-protein interaction

BTH kits (EUROMEDEX, France) and cloning strategies were employed according to established protocols[52,53]. The recombinant pKT25 and pUT18C plasmids were then transformed into BTH101 strain and plated on LB agar containing 1 mM IPTG in the presence of 100 μg mL$^{-1}$

ampicillin, 50 μg mL$^{-1}$ kanamycin and 50 μg mL$^{-1}$ 5-bromo-4-chloro-indolyl-β-D-galactopyranoside (X-Gal). After incubation at 30 °C for 24 h, positive interactions were indicated by blue colonies and quantified by β-galactosidase assay (according to the instructions)[54]. In the study, ZIP1 and ZIP2 were a pair of functionally complementary strong interaction plasmids.

## Static biofilm phenotype assay

The bacterial suspension was adjusted to a consistent OD$_{600}$, and 2 mL of LB liquid medium was added to borosilicate glass tubes. The original suspension was then diluted 1:100 and 20 μL of the culture was added to each tube. 1 mM IPTG and the appropriate antibiotics were also added, with each group being repeated three times. In experiments investigating the effects of extracellular environmental factors on bacterial biofilms, different compounds were added to the culture medium: iron depletion, BIP medium (200 μM 2,2'-bipyridyl), iron supplementation (100 μM FeCl$_3$), and Heme supplementation (40 μM). After incubation at 30 °C for 18 h, the tubes were removed, and the culture was poured out. The tubes were rinsed with water, air-dried, and then stained with 3 mL of 0.1% crystal violet solution for 15 min. The tubes were rinsed three times with water and air-dried before being photographed for documentation. For biofilm quantification, 2.5 mL of 95% ethanol was added to dissolve the stain, the tubes were sealed and placed on a shaker at room temperature for 12 h. The OD$_{595}$ was measured and recorded.

## Motility phenotype assay

15 mL of swimming medium (Peptone: 10 g L$^{-1}$, Sodium Chloride: 10 g L$^{-1}$, Agarose: 3 g L$^{-1}$), swarming medium (Bouillon: 8 g L$^{-1}$, Agarose: 10 g L$^{-1}$), twitching medium (Peptone: 10 g L$^{-1}$, Agarose: 10 g L$^{-1}$, Sodium Chloride: 5 g L$^{-1}$), and Congo red (Peptone: 10 g L$^{-1}$, Agarose: 3 g L$^{-1}$, Coomassie Brilliant Blue G-250: 20 mg mL$^{-1}$, Congo red: 40 mg mL$^{-1}$) were added to a 9 cm disposable petri dish along with appropriate antibiotics and IPTG. The petri dish was left at room temperature for 2 h. The bacterial suspension was adjusted to an OD$_{600}$ of 0.6 and diluted at 1:100. 0.6 μL of the suspension was pipetted onto the petri dish and incubated at 30 °C for 12 h. The petri dish was then removed and photographed, and the radius of the bacterial colony was measured. Each experiment was repeated three times.

## Growth curve assay

Bacterial monoclone was subjected to overnight culture, resulting in the successful preparation of specific seed solution. Followed by adjusting the corresponding seed solution to an OD$_{600}$ of 1.0, which was then inoculated at a 1:1,000 dilution into two media: M9 and M9 + Fe$^{3+}$ (100 μM FeCl$_3$) with the OD600 values monitored. And both media contained 1 mM IPTG, 200 μg mL$^{-1}$ carbenicillin and 100 μg mL$^{-1}$ tetracycline.

## Luminescence screening assays

The luminescence of lux-based reporters in cells grown in liquid culture was quantified using a Synergy 2 plate reader (BioTek) as previously reported[55]. In brief, reporter strains were cultured overnight and then diluted to an OD$_{600}$ of 0.2 with fresh LB medium. After incubation for an additional 2 h at 37 °C with shaking, the cultures were further diluted by 1:20 with 100 μL of fresh LB and transferred to a black 96-well plate with a transparent bottom. Sterilized mineral oil (50 μL) was added to each well. Luminescence intensity and bacterial growth (OD600) were monitored every 30 min for a total of 24 h using the Synergy 2 plate reader.

## Pull-down assay for protein interaction validation

Recombinant pMMb67H-ImsP-eGFP and pME6032-ImsP-Flag expression vectors were constructed and subsequently electroporated into PAO1. Monoclonal colony was selected from a LB solid plates

containing 200 μg mL$^{-1}$ carbenicillin and 100 μg ml$^{-1}$ tetracycline, followed by being inoculated to liquid LB at 37 °C (220 rpm, 6 h). 50 mL of post-induced bacterial culture was collected (12,000 g, 5 min) and resuspended with 1 mL of buffer (50 mM Tris-HCl pH 8.0, 150 mM NaCl, 0.1% (v/v) Triton X-100). Cells were cracked using an ultrasonic disruptor and centrifuged (12,000 g, 5 min) to collect the supernatant. 50 μL of Flag beads were added and incubated for 2 h on a rotator at 4 °C. The beads were then washed three times. The retained proteins were detected by western blotting after SDS-PAGE. In our experiments, we used RNA Polymerase (RNAP) as a loading control. We ensured that all blots and gels were accompanied by the locations of molecular weight/size markers. We also provided uncropped and unprocessed scans of the most important blots, which have been added to the Source Data file. The eGFP, Flag, and RNAP antibodies, along with the secondary antibodies, are all meticulously prepared at a dilution ratio of 1:2000 to ensure optimal experimental conditions.

## Flow cell biofilm

The target bacterial strain was grown in liquid LB medium supplemented with 100 μg ml$^{-1}$ tetracycline at 37 °C and 220 rpm for 14 h. The bacterial culture was then adjusted to a uniform optical density using LB medium and 300 μL of the culture was transferred to a flow chamber. A control flow chamber was also set up with 300 μL of LB medium. The flow chamber was connected to a single-use channel made of plastic and silicone tubing arranged for unidirectional flow. Sterilized LB medium containing 1 mM IPTG was introduced into the chamber at a rate of 3 mL h$^{-1}$ via a peristaltic pump at 30 °C for 48 h. For staining, 300 μL of SYTO9 (Thermo Fisher Scientific SYTO9SYTO™ 9, USA) was slowly added to each channel and incubated in the dark at room temperature for 30 min. The biofilm structure was visualized using a confocal laser scanning microscope (Zeiss LSM 900, Germany) and four random images were captured from each flow chamber. Z-stack images were generated at 1.4 μm intervals with the number of slices chosen to cover the entire microcolony in each image area. The excitation wavelength was set to 483 nm and the emission wavelength to 500 nm for SYTO9 detection, with image stacks acquired using a 10X objective lens. Image analysis was performed using COMSTAT2 software to quantify bio-volume (biomass volume divided by substrate area; μm$^3$ μm$^{-2}$) and maximum biofilm thickness (μm), providing estimates for biomass and spatial size of the biofilm in the flow cells.

## High-performance liquid chromatography

The protein sample was incubated with 200 μM GTP (SIGMA, Germany) in buffer (50 mM Tris-HCl, pH 8.0, 150 mM NaCl, 0.025% DDM) at 37 °C for 2 h. 100 μm solution of C-di-GMP was used as a standard. In the necessary reactions, 200 μm ml$^{-1}$ of FeCl$_3$ was added to the protein and incubated at room temperature for 6 h prior to the addition of GTP for the reaction. The reaction was then terminated by heating at 85 °C for 5 min. Detection was performed on a 1260 Infinity II instrument using a ZORBAX SB-C18 column (30 ×100 mm, 5 μm; Agilent, USA) at a flow rate of 0.3 ml min$^{-1}$ with an injection volume of 2.5 μL per run. The mobile phase was methanol and the stationary phase was water containing 1% ammonium acetate. Data analysis was performed using the Agilent Academic Data Analysis Software Program.

## Metal content determination in purified proteins

Metal ions (Mn$^{2+}$ and Fe$^{3+}$) in CHASE4 were measured by Inductively Coupled Plasma Mass Spectrometry (ICP-MS) using an Agilent 7700x system. Proteins expressed in two media: M9 only and M9 + Fe$^{3+}$ (100 μM) were required for element analysis. FeCl$_3$ (100 mM) was added before lysing the bacteria, followed by vortexing for homogeneity, then let stand at 4 °C for 2 h. The protein (1 mg) was dissolved in 5 mL of 10% hydrochloric acid for 2 h. Standard curves for different metal ion concentrations (0, 1, 5, 10, 50, and 100 ppb) were prepared using a multi-element standard solution for ICP-MS (90243, Sigma-Aldrich, USA). The purified protein was measured after being collected by Size Exclusion Chromatography (SEC). Each protein sample was measured in triplicate and the results were calculated based on the standard curves.

## Liquid chromatography-mass spectrometry

10 mL bacterial suspension induced with 1 mM IPTG was centrifuged at 8000 g and 4 °C for 2 min. The pellet was transferred and resuspended in 5 mL of PBS. The suspension was then centrifuged again at 8000 g and 4 °C for 2 min and the pellet was washed twice. The pellet was resuspended in 5 mL of extraction buffer (methanol: acetonitrile: water = 2:2:1). The suspension was flash frozen in liquid nitrogen and thawed before vortexing until no visible bacteria remained. The suspension was then sonicated for 5 min at low temperature (repeated three times). The resulting liquid was centrifuged at 12,000 g and 4 °C for 30 min and the supernatant was freeze-dried to a powder using a CentriVap Benchtop Vacuum Concentrator at 4 °C. The powder was resuspended in 200 μL of extraction buffer and centrifuged at 12,000 g and 4 °C for 10 min. The supernatant was transferred to a mass spectrometry sample tube for analysis. Identification and relative quantification of c-di-GMP were performed using a triple quadrupole mass spectrometer (Q-Exactive, ThermoFisher Scientific, USA). Standards were dissolved in water and separated on a C18 column using a binary pump system with solvent A consisting of water containing 0.1% (v/v) formic acid and eluent B consisting of acetonitrile containing 0.1% (v/v) formic acid. The gradient started at 10% eluent B and was held for 10 min at a flow rate of 0.3 m min$^{-1}$. The column temperature was maintained at 30 °C and the autosampler temperature was set at 4 °C. Data were analyzed using Xcalibur (version 4.0) Image J (version 1.5.4) and Trace Finder (version 4.1).

## Phylogenetic Analyses

Protein sequence similarity searches were carried out with the BLASTP program from the National Center for Biotechnology Information. Phylogenetic and molecular evolutionary analyses were conducted using neighbor-joining method as implemented in MEGA (version 11) with a bootstrap value of 1000[56].

## Protein expression and purification

For membrane proteins, *E. coli* BL21 (DE3) cells harboring the expression plasmid were grown in LB medium supplemented with appropriate antibiotics at 37 °C. Protein expression was induced by adding 0.1 mM IPTG at 16 °C for 20 h after the OD600 reached 0.8. The cells were collected by centrifugation at 4000 g for 20 min and lysed in lysis buffer (30 mM HEPES pH 7.5, 150 mM NaCl). The lysates were centrifuged at 12,000 g at 4 °C for 30 min to remove cell debris, supernatants were further centrifuged at 100,000 g at 4 °C for 60 min to collect membrane. The membrane was re-suspended and dissolved in lysis buffer containing 1% (v/v) n-dodecyl-β-D-maltoside (DDM) at 4 °C for 2 h and further centrifuged at 20,000 g at 4 °C for 30 min to remove insoluble fractions. The supernatant was applied to Strep-Tactin® Sepharose beads (IBA Lifesciences, Germany). After washing with 20 column volumes (CV) of wash buffer (30 mM HEPES pH 7.5, 150 mM NaCl and 0.01% LMNG), the recombinant proteins were eluted using wash buffer with 10 mM D-biotin (IBA Lifesciences, Germany) added. The eluent was further purified through Superose® 6 increased 10/300 GL (GE Healthcare Life Sciences, USA). Similar methods are employed for other membrane proteins. In the case of the CHASE4 protein and its site-directed mutant variants, the cells were lysed using a lysis buffer (50 mM HEPES pH 8.0, 300 mM NaCl). The lysates were then centrifuged at 12,000 g at 4 °C for 30 minutes to remove cell debris. Subsequently, the supernatant was applied to a 5 mL HisTrap HP column (Cytiva). The target protein was finally eluted using an AKTA purifier (GE Healthcare) with a

wash buffer composed of 20 mM HEPES (pH 8.0), 300 mM NaCl, and 300 mM imidazole.

## Microscale thermophoresis (MST)

The PA2072[His] protein (100 nM) was subjected to fluorescent labeling using a His-tag labeling kit (Nano Temper, MO-L008), following the instructions provided in the user manual. Upon completion of the labeling process, 10 μL of the labeled PA2072[His] protein was combined with 10 μL of PBST buffer, PA1851 protein (200 nM) was then added to the mixture, which was then serially diluted. Microthermophoresis was performed using a NanoTemper monolith NT.115 instrument, operating at 100% LED power and high MST power. The data procured were subsequently analyzed using MO. Offinity Analysis (X86).

## Cryo-EM grid preparation and microscopy

After glow-discharging for 30 seconds with Plasma cleaner, 3 μL of samples (10 mg/ml) was applied to the carbon side of Quantifoil R 0.6/1 gold 200 mesh grid grids, blotted for 4.0 seconds without blot force, and plunge-frozen in liquid ethane using Vitrobot (Thermo Fisher Scientific, USA). Cryo-EM grids were imaged on a Thermo Fisher Scientific 300 kV TEM Titan Krios using Gatan K3 direct electron detector. Raw movies were collected in super-resolution mode at a magnification of 130,000, with a super-resolution pixel size of 0.334 Å. The data was stored in 32-frame gain normalized stacks with a total dose of 55 $e^-/Å^2$. The detailed information of data collection was summarized in table S1.

## Cryo-EM image processing and model building

A total of 114,509 movie stacks were imported into Cryosparc v3.3.2[57]. After Motion-Correction with bin 2 and CTF Estimation, micrographs (pixel size 0.668 Å) with CTF-estimated maximum resolution better than 4 Å were selected to pick particles using Template Picker. After one round of 2D classification, a set of 629,049 particles were subjected to two rounds of Heterogeneous Refinement using one good template (from Ab-Initio Reconstruction) and three random templates. The good class was refined by one round of Homogeneous Refinement without symmetry, and further by one round of Non-uniform Refinement with symmetry C2 and CTF refinement. The final map was sharped with Phenix_auto_sharpen[58]. Figures representing the map features were prepared with UCSF Chimera[59]. The details of the data collection and processing refer to Supplementary Table 1 and Fig. 5a–f. A predicted structure from AlphaFold2 was used as a template for model building and refinement. The model was further improved through cycles of real-space refinement (with Ramachandran restraints and secondary structure restraints) in Phenix and following manual corrections by Coot[58,60]. The refinement statistics of the model are summarized in Supplementary Table 1.

## Protein crystallization X-ray diffraction data collection and processing

The CDS fragment of recombinant IsmP was constructed to pET-28a vector and further expressed in BL21 strain. The purified IsmP dissolved in buffer containing 20 mM HEPES (pH 8.0), 100 mM NaCl was concentrated to 10 mg/mL using ultrafiltration device. Crystallization screening was performed at 289.15 K applying the sitting-drop vapor diffusion method. The crystals used for data collection were picked from the condition consisting of 0.2 M sodium acetate lithium, 20% (v/v) PEG3350, 0.1 M Bis-Tris propane, pH 7.9. Followed by instantly flash-frozen in liquid nitrogen with a mixture of 20% (v/v) glycerol and reservoir as cryoprotectant.

Crystal diffraction experiment was conducted using single-wavelength small angle oscillation method at beamline 18 u located in Shanghai Synchrotron Radiation Facility (SSRF). All diffraction frames were integrated with XDS&XDSGUI[61]. Followed by data scaling using Aimless with ~5% unique reflections used for calculating $R_{free}$[62].

The analysis by Xtriage suggested that there was slight translational non-crystallographic symmetry (tNCS) found in the merged dataset, which could make little effect on subsequent phasing step, however, easily lead to a higher R factor than expected. And the structure factor phase was searched by PHASER using molecular replacement method (MR) with the model predicted by AlphaFold2 as a search template[63–65]. Autobuild was used to complete the initial model. Coot and Phenix.refine were further used to refine the atomic coordinate with Molprobity as an evaluation tool[58,66]. Crystallographic statistical indicators referring to data collection and refinement are given in Supplementary Table 2.

## Modelling for ImcA-IsmP2-ImcA hetero-tetramer and Fe³⁺-bound ImcA homo-dimer based on in silico analysis

The predicted models of full-length ImcA and IsmP deposited in AlphaFold Protein Structure Database were used as starting model. HDOCK sever was utilized to analyze the potential interaction mode between ImcA and IsmP, followed by two rounds of molecular dynamics simulation, simulated annealing with centroid restraints and water refinement included as implemented at the HADDOCK2.4 server[67,68]. The final ImcA-IsmP hetero-tetramer candidate docking model was validated by PISA [https://www.ebi.ac.uk/pdbe/pisa/]. Besides, HADDOCK2.4 server was utilized to predict and refine the potential binding pose combined with ColabMetal3D[69].

## Statistics and Reproducibility

All experiments were repeated at least 3 times, and statistics were based on biological replicates. Detailed information about replicates and statistical analyses for each experiment is provided in the figure legends and the Source data file. All analyses were performed using GraphPad Prism. Differences were considered significant when the $P$ value was <0.05. Experiments were not randomized; and investigators were not blinded to allocation during experiments and outcome assessment.

## Reporting summary

Further information on research design is available in the Nature Portfolio Reporting Summary linked to this article.

## Data availability

Source data are provided with this paper. The cryo-EM density maps for ImcA have been deposited in Electron Microscopy Data Bank (EMDB) under accession code EMD-37444. The corresponding atomic coordinates have been deposited in the Protein Data Bank (PDB) under accession code 8WCN. The atomic coordinates of IsmP homodimer crystal structure have been deposited in PDB under accession code 8WCT. Prediction models used as starting point for modelling the IsmP-ImcA oligomer can be found in AlphaFold Protein Structure Database under accession codes Q9I243 for IsmP and Q9I2P4 for ImcA. Source data are provided with this paper.

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

## Acknowledgements

We thank the staff at Southern University of Science and Technology (SUSTech) Cryo-EM Center for assistance in data collection on the SUSTech cryo-electron microscope. We extend our gratitude to Researcher Jiahai Zhou from the Shenzhen Institutes of Advanced Technology, Chinese Academy of Sciences, for his invaluable assistance in protein crystallization. We acknowledge the support from the National Key Research and Development Program of China (2022YFC2304400, 2022YFC2304401 to H.L.), the Shenzhen Science and Technology Program (20231120104808001 to H.L.), and the National Natural Science Foundation of China (32170188 to H.L)

## Author contributions

X.L.Z. conceived the idea, designed the experiments, conducted the experiments, and wrote the manuscript. K.Z. and C.W. analyzed the data from cryo-electron microscopy and protein crystals, and contributed to the writing of the manuscript. Q.F., X.T. and X.Z. helped with the characterization and data analysis. K.W. and Y.F. participated in the discussions and provided financial support. H.L. conceived the idea, designed the experiments, wrote the manuscript and provided financial support.

## Competing interests

The authors declare no competing interests.
