## [Peer Review File · Nature Communications]

A c-di-GMP signaling module controls responses to iron in
Pseudomonas aeruginosaReviewer #1 (Remarks to the Author):

See attached

Reviewer #1 Attachment on the following page

Review of A local signaling switch controls *Pseudomonas aeruginosa* behaviors in response to iron by Zhan *et al.*

Summary: In this work the authors present new insights into second messenger signaling, including two structures of previously (structurally) unknown proteins. While the structural characterization reports novel findings of potentially important enzymes, there are major flaws that make it difficult to appreciate what impact these findings have and to determine if the data truly supports the conclusions made. Overall, the manuscript is relatively poorly written and presented in such a way that is sometimes confusing and, at other times, contradictory. There appears to be good science here but also some holes in the reasoning. Properly organized, and with the data more clearly communicated, this work has the potential to be of high impact, but certainly not as currently presented. More details are provided below.

Some of the major issues identified:

- Some of the experiments and discussion surrounding the iron binding were confusing and appear to be lacking appropriate controls. For example, why did the authors use only the Fe³⁺ ion for their experiments (or only report this data) and use divalent cations (Mn²⁺ and Mg²⁺) as a control? Why not include data for Fe²⁺ or other cations as controls? The authors identify a putative site for iron binding in the CHASE4 domain (and model iron into this site), however it is unclear if the appropriate structural architecture to coordinate the ion is present at this site. If it is, the coordination to the ion should be shown and, if not, some rationale should be presented for why the side chains do not adopt the ideal geometry. In addition, some mention of Fe²⁺ vs Fe³⁺ homeostasis and/or metabolism in *p. aeruginosa* would help to strengthen the story and answer some additional questions. The data provided for heme binding is, perhaps, interesting but is incomplete. How would heme bind to the protein seeing as it is very structurally distinct from an Fe³⁺ ion? Can you rule out that it is not just iron leaching from the heme that is responsible for the results? The ITC data presented for Heme is not of great quality and would benefit from the presentation of appropriate controls.
- The authors propose a model for how lsmP and lmcA putatively interact and a mechanism for how this interaction is regulated by iron binding. Beyond just being highly speculative, the data does not really seem to support these models except for the evidence for some role for iron regulating their, potentially, coordinated functions. While the B2H data may suggest an interaction, and the docking provides a possible binding pose, the structures themselves seem to indicate otherwise. The lmcA structure presented shows a relatively tight dimer that would have to dissociate to accommodate lsmP as proposed. No evidence is provided, nor is a means for how this would occur proposed. The proposed reorganization of lsmP upon iron binding is presumably based upon some observations of presumed flexibility (from B-factors and other observations), which are not compelling. The site directed mutagenesis experiments described on pg. 13 were perhaps suggestive but, to me, were not convincing evidence that this interaction occurs as proposed. More structurally conserved mutants that showed similar results would provide a stronger case, as would measurement of the interaction using an independent biophysical approach (since the authors reported purifying both of the proteins for structural analysis, this should be reasonably trivial).
- In quite a few places throughout the manuscript, insufficient details were provided to be able to properly interpret the results including in the figure captions. Some examples include: how the CHASE4 domain was identified/chosen and which residues were included for the protein expression and structure; how the mutagenesis experiments were designed and carried out (paragraph 2, page 12, in particular); many of the figure

captions provide insufficient details including data presented in Fig. 2a, details about the maps shown in Fig. 4, and details about the gels shown in extended data Fig. 4 (what is in each lane), among others.

- Some of the experimental details are missing. Examples include how the CHASE4 domain protein was expressed and purified, how the proteins were expressed and treated for the Mn or Fe supplementation experiments (ie was it just the media that was supplemented and/or were the metal salts added to purified proteins?); native page data was reported (extended data Fig. 4) but not described etc.

Some of (not a complete list) of minor issues identified:

- Throughout the manuscript are a great number of typographical, formatting and grammar errors. While not directly related to the science, this does, at times, make it difficult for the reader to understand the point(s) the authors are trying to make. Some examples are provided:
 - o p2, l32: it is not clear what is meant by “that besides the catalytic sites”
 - o the authors need to define ImcA and IsmP earlier; ie perhaps include a description like what is given in lines 114-117 earlier in the text
 - o l102: this would benefit from a brief introduction to the assay design for what “wrinkled colonies” represent
 - o l202: the reference to Fig. 2d is incorrect
 - o l260: “non-crystallographic” is not correct since this is not a crystal structure
 - o ll304-305: shouldn’t use “Obviously” and “presumably” together this way
 - o l378: the font appears to be different here
 - o l571: MnCl₂ and FeCl₃ are not metal ions
 - o l643: should read “sharpened by Phenix_auto_sharpen”
 - o Extended data Fig. 4 and 8. Captions – text at end is somehow different
- There is disagreement between the data shown in extended Table 2 and what is reported in the text (resolution and space group, in particular)
- The methods appear to describe the expression of full length IsmP, but the structure solved was only the CHASE domain
- It would be good to include the protein only control (eg. ImcA without GTP added) for the HPLC experiment, to see what may copurify
- Why are there multiple peaks for the GTP standard in HPLC?
- The representations of the GMPCPP binding to IMCA shown in Fig. 4 do not do a good job of showing how well the density corresponds to the putative ligand. I would suggest additional figures (perhaps in the extended data) showing difference/composite omit maps from a couple of different angles to provide convincing evidence of the fits.
- Some of the text in figures is very small, particularly in the extended data figures. For example, the labeling of the ITC data in extended data Fig. 4.

Reviewer #2 (Remarks to the Author):

This manuscript by Zhan et. al. describes the discovery of a novel iron-sensing c-di-GMP signaling module consisting of two membrane proteins the authors designate *IsmP*, which senses iron, and *ImcA*, a diguanylate cyclase (DGC) that synthesizes c-di-GMP. The authors provide evidence that these two proteins interact and at low concentrations of iron, *IsmP* inhibits the DGC activity of *ImcA*. Iron binding to the periplasmic CHASE4 domain of *IsmP* then causes it to dissociate with *ImcA*, leading to increase c-di-GMP and biofilm formation. I think the experiments are well done and clearly described, and the authors present significant *in vivo* and *in vitro* data to support this model. However, I do have some concerns about the physiological relevance of this iron responsive module as none of the relevant phenotypes are shown in a WT cell and the delta *ismP* mutant does not have a hyper-biofilm forming phenotype. I also think the authors oversell the evidence that this pathway exhibits signaling specificity. However, given the paucity of known environmental cues that regulate c-di-GMP signaling, and the mechanism by which they are sensed, this work will be of significant interest to the field and make a significant impact, so overall I am enthusiastic about it. Specific comments are:

1. One significant concern I have with the manuscript as I highlight below in points 3 and 7 is that the interesting phenotypes are only observed when *imcA* is overexpressed in the *ismP* deletion mutant. For example, in Fig. 2d biofilms of the WT strain are not impacted by changes to iron. This brings into question the physiological relevancy of the results. The model in Fig. 6 then is not entirely accurate as such impacts were not shown in a WT cell. The authors should address this discrepancy in this discussion and argue for the relevance of the results.
2. The discussion at multiple points suggests that their results demonstrate the *ImcA* and *IsmP* module shows localized signaling specificity, but I would disagree with this statement as they provide no evidence for such localized signaling. They first make this claim in Lines 415-416 stating "Therefore, *IsmP-ImcA* module meets a local c-di-GMP signaling model.", but this seems to be based on similarities to the *NosP-NahK* signal. In my view, their data suggests a generalized signaling module. In order to show localized or specific signaling, there needs to be a significant impact of the activity of a DGC or PDE on a c-di-GMP-regulated process in the absence of global changes to c-di-GMP. However, the author's data, particularly in Fig. 1, show that biofilm formation and colony morphology changes are only apparent in strains where the overall concentration of c-di-GMP increases. If there are other pieces of evidence to suggest specific signaling, the authors need to do a better job of making their case. Otherwise, any claim for specific signaling should be removed.
3. Extended Data 1: If *IsmP* is inhibiting the DGC activity of *ImcA*, then why does the delta *ismP* mutant exhibit reduced biofilm? In other words, why do the authors need to overexpress *IsmA* to see a phenotype?
4. Lines 96-117-This paragraph is a bit confusing as the authors switch between using PA2072 and *IsmP* without first defining that *IsmP* is encoded by this gene. I suggest they use the PA2072 designation until line 115 when they now define it as *IsmP*.
5. Line 137-This should be Extended Data Fig. 3, and I do not see any data to show wrinkling on CR plates.
6. Line 142-Again, there is no wrinkling colony morphology in Extended Data Fig. 3. I think the authors are referring to Fig. 1d.
7. Fig. 1e,f-Similar to point 3 above, why doesn't the delta *ismP* mutant exhibit enhanced c-di-GMP levels as *imcA* would be derepressed. Or is *IsmA* not expressed in this mutant?
8. Line 188-As *K_d* refers to the concentration at which half of the protein is bound to the small molecule and is the inverse of the association constant, PA0847 has a weaker binding affinity for Fe⁺³ at 16 μ M compared with *IsmP* CHASE domain at 6.9 μ M.
9. Fig. 2d-This might be my pdf file, but the text on this figure is not sharp and difficult to read. It is also worth highlighting what conditions were used for this experiment. For example, the authors should state that BIP is an iron chelator.
10. Line 456-The authors should reference the plasmids and primers table in this section.

Reviewer #3 (Remarks to the Author):

The authors identify PA2072 as a gene that, when either overexpressed or deleted, led to a decreased biofilm production. The authors performed bacterial two hybrid screens and identified a number of proteins involved in c-di-GMP synthesis and degradation that interact with PA2072 including PA1851. The authors show that overexpression of PA1851 (renamed *imcA*) in a strain lacking PA2072 (renamed *ismP*) led to increased biofilm. The author perform biochemistry and show that *ImcA* can synthesize c-di-GMP in vitro, but this activity is inhibited by the addition of *IsmP* or the *IsmP* Chase4-PASPAC domain. The inhibition by *IsmP* can be reversed by the addition of Fe^{3+} . The authors solved the crystal structure of the CHASE4 domain of *IsmP* and modeled in the Fe^{3+} . Alteration of the predicted residues that bind Fe^{3+} prevented binding to iron and addition of Fe^{3+} no longer reversed inhibition of *ImcA*. The authors solved the EM structure of *ImcA* with a non-hydrolysable GTP analog, which was similar to the AlphaFold model. The authors modeled *IsmP* and *ImcA* which led to a number of residues predicted to mediate the interaction. Site directed mutants of those residues prevented inhibition of *ImcA* diguanylate cyclase activity by *IsmP*.

Major comments:

1. The pull-down is not very convincing. If the authors have the purified proteins, can they be mixed and analyzed by SEC as shown in Extended Data Fig. 4F. This would provide better support for the interaction between these two proteins than the docking model. This should also be done with Fe^{3+} to show the complex dissociates.
2. The authors' model suggest that inhibition is based on stoichiometric interaction between *IsmP* and *ImcA*. This raises a number of questions. Shouldn't overexpression of *ImcA* overcome inhibition by *IsmP*, but this is not the case in Figure 2D (PAO1/EV vs PAO1/p-*imcA* in LB). How much *IsmP* protein is in PAO1 strain? How much *ImcA* is made when expressed from the plasmid?
3. Figure 2D shows that addition of iron to PAO1/EV does not have any effect on biomass of the biofilm. This raises a follow-up to the above point is how much *ImcA* is in PAO1? Is it just not expressed?
4. The authors propose Fe^{3+} binding in the micromolar range. The authors should discuss when is amount of Fe^{3+} likely to be possible in the environment or during infection.
5. While a writing issue, this is a major problem since the authors starts using *IsmP* (line 100) and *ImcA* (line 110) prior to naming the two proteins in line 115-117. Please introduce the name before using it. Prior to the renaming, just use the PA gene number. Please stop using the PA gene number in Material and Methods and Extended Data since you are renaming them.

Minor issues:

1. For the bacterial two hybrid assays, showing the bait plasmid with an empty prey plasmid would be a better negative control than two empty vectors.
2. Abstract – several statements are overgeneralizations. Please be more specific. Line 23 – “However, environmental signals controlling the intracellular c-di-GMP levels still remain enigmatic.” and Line 36 “CHASE4 domain directly senses the environmental signals” The authors should specify iron since many environmental signals are known for enzymes regulating c-di-GMP.
3. Line 32 – please clarify meaning of “unveiled a unique conformation that besides the catalytic site”
4. Replace reference #5 in line 50. This is not an appropriate reference for this statement.
5. Add reference for PdeR-DgcM-MlrA in line 63.
6. The biochemical data in Figure 2A needs to be clarified and quantified.
A. The amount of proteins used in diguanylate cyclase assay is not mentioned for Fig 2A. This

makes it nearly impossible to understand the data being presented.

B. Different ratios of the ImcA and IsmP should be used to show that there is a stoichiometry for the observed inhibition.

C. The authors states in line 165 that "However, GTP remained the primary compound in the reaction..." The figure shows that c-di-GMP is being made, so quantification of the data is needed.

7. Authors should test IsmP Chase4-PASPAC to inhibit Δ ismP/p-imcA. This data should go into Figure 3I.

8. Avoid using the word – obvious or obviously. Let the readers make that determination.

9. The title is "A local signaling switch controls *Pseudomonas aeruginosa* behaviors in response to iron". Why is IsmP and ImcA localized signaling? Is it just because they physically interact? Local signaling suggest that the signaling molecule is signaling locally rather than globally, so the system described doesn't seem to show that. Perhaps change title to "A c-di-GMP signaling module controls *Pseudomonas aeruginosa* behaviors in response to iron". This is more accurate to the proposed model.

Dear editor and reviewers:

Thank you for your letter and for the reviewers' comments concerning our manuscript entitled "A local signaling switch controls *Pseudomonas aeruginosa* behaviors in response to iron" (ID: NCOMMS-23-51914-T). We have now performed a series of additional experiments and incorporated relevant data into the revised manuscript. Herein we have carefully addressed concerns and suggestions raised by the editor and reviewers and provide our point-by-point responses below.

Reviewer #1 (Remarks to the Author):

Summary: In this work the authors present new insights into second messenger signaling, including two structures of previously (structurally) unknown proteins. While the structural characterization reports novel findings of potentially important enzymes, there are major flaws that make it difficult to appreciate what impact these findings have and to determine if the data truly supports the conclusions made. Overall, the manuscript is relatively poorly written and presented in such a way that is sometimes confusing and, at other times, contradictory. There appears to be good science here but also some holes in the reasoning. Properly organized, and with the data more clearly communicated, this work has the potential to be of high impact, but certainly not as currently presented. More details are provided below.

Some of the major issues identified:

1. Some of the experiments and discussion surrounding the iron binding were confusing and appear to be lacking appropriate controls. For example, why did the authors use only the Fe^{3+} ion for their experiments (or only report this data) and use divalent cations (Mn^{2+} and Mg^{2+}) as a control? Why not include data for Fe^{2+} or other cations as controls? The authors identify a putative site for iron binding in the CHASE4 domain (and model iron into this site), however it is unclear if the appropriate structural architecture to coordinate the ion is present at this site. If it is, the coordination to the ion should be shown and, if not, some rationale should be presented for why the side chains do not adopt the ideal geometry. In addition, some mention of Fe^{2+} vs Fe^{3+} homeostasis and/or metabolism in *p. aeruginosa* would help to strengthen the story and answer some additional questions. The data provided for heme binding is, perhaps, interesting but is incomplete. How would heme bind to the protein seeing as it is very structurally distinct from an Fe^{3+} ion? Can you rule out that it is not just iron leaching from the heme that is responsible for the results? The ITC data presented for Heme is not of great quality and would benefit from the presentation of appropriate controls.

Response: We sincerely thank the Reviewer's positive evaluation for our manuscript. The constructive comments and suggestions that have helped us improve the quantity of our manuscript. We provided the point-by-point responses below.

a. Some of the experiments and discussion surrounding the iron binding were confusing and appear to be lacking appropriate controls. For example, why did the authors use only the Fe^{3+} ion for their experiments (or only report this data) and use divalent cations (Mn^{2+} and Mg^{2+}) as a control? Why not include data for Fe^{2+} or other cations as controls?

Response: Thanks for the good suggestion. We have repeated ITC assays and used Fe^{2+} as a control. The experimental results showed that Fe^{2+} does not bind with the CHASE4 protein (**Supplementary Fig. 4b**). We initially chose Mn^{2+} and Mg^{2+} as controls because they are likely to affect the catalytic activity of c-di-GMP metabolic enzymes.

b. The authors identify a putative site for iron binding in the CHASE4 domain (and model iron into this site), however it is unclear if the appropriate structural architecture to coordinate the ion is present at this site. If it is, the coordination to the ion should be shown and, if not, some rationale should be presented for why the side chains do not adopt the ideal geometry.

Response: Thanks for the suggestive comment. As described in the corresponding “Materials and Method” part, the Fe^{3+} was modelled into potential Fe^{3+} binding pocket of $\text{IsmP}^{\text{CHASE4}}$ using molecular docking. However, it seems that molecular docking could not further confirm whether a metal atom, including Fe^{3+} , will be coordinated by residue sidechains from receptor protein, even combined with simple dynamics (it may need much more complex molecular dynamics simulation to make it). In addition, it is difficult to determine whether water molecules around the binding pocket refer to the coordination with Fe^{3+} . Most importantly, what we want to demonstrate is which residues located in the potential binding pocket may interact with the metal atom (Fe^{3+}) after the fact that Fe^{3+} could bind to $\text{IsmP}^{\text{CHASE4}}$ was verified by biochemical and biophysical experiments (ICP-MS, ITC). And it should make little effect on the conclusion that iron could bind to $\text{IsmP}^{\text{CHASE4}}$ without confirming whether the iron will be coordinated. Actually, that is somewhat why we didn’t present more details on the interaction, like distance between bonding atoms. As for the side chains that did not adopt the ideal geometry, the pocket presented in the Fig.3g was actually refined using refinement module as implemented on Haddock2.4 web server. And we are afraid to further adjust the side chains leading to worse pose. And the fig. 3g has been renewed to present details about the orientation changes of side chains after refinements compared to the orientations of those in corresponding residues in the crystal apo structure.

c. In addition, some mention of Fe^{2+} vs Fe^{3+} homeostasis and/or metabolism in *P. aeruginosa* would help to strengthen the story and answer some additional questions.

Response: Thanks for the good suggestion. We have discussed the homeostasis and/or metabolism of Fe^{2+} and Fe^{3+} in *P. aeruginosa* as following: “It has been shown that *P. aeruginosa* acquires trivalent iron and heme from the blood via various carriers. Inside the cell, Fe^{3+} is reduced to Fe^{2+} , participating in biochemical reactions and physiological processes. Moreover, the trivalent iron concentration in the respiratory tract of cystic fibrosis patients exceeds that of healthy individuals, surpassing 8 μM . Our data will expand understanding of how iron ions enhance the infectivity of *P. aeruginosa* during infection through the induction of CHASE4 (lines 380-384)”.

d. The data provided for heme binding is, perhaps, interesting but is incomplete. How would heme bind to the protein seeing as it is very structurally distinct from an Fe^{3+}

ion? Can you rule out that it is not just iron leaching from the heme that is responsible for the results? The ITC data presented for Heme is not of great quality and would benefit from the presentation of appropriate controls.

Response: We agree with the reviewer that heme binding data is incomplete. The heme reagent, as per the product manual, can only be dissolved with a strong alkali. We attempted to adjust the pH value but was unsuccessful due to the high background heat, which precludes the ITC experiment for testing the interaction between heme and CHASE4. Therefore, we only measured the effects of heme on bacterial growth and biofilm phenotype. Simultaneously, the ITC experimental results indicate that divalent iron does not bind with CHASE4, which rules out the possibility of divalent iron ions dissociating from heme. The interaction between iron and CHASE4 needs to be further investigated.

2. The authors propose a model for how IsmP and ImcA putatively interact and a mechanism for how this interaction is regulated by iron binding. Beyond just being highly speculative, the data does not really seem to support these models except for the evidence for some role for iron regulating their, potentially, coordinated functions. While the B2H data may suggest an interaction, and the docking provides a possible binding pose, the structures themselves seem to indicate otherwise. The ImcA structure presented shows a relatively tight dimer that would have to dissociate to accommodate IsmP as proposed. No evidence is provided, nor is a means for how this would occur proposed. The proposed reorganization of IsmP upon iron binding is presumably based upon some observations of presumed flexibility (from B-factors and other observations), which are not compelling. The site directed mutagenesis experiments described on pg. 13 were perhaps suggestive but, to me, were not convincing evidence that this interaction occurs as proposed. More structurally conserved mutants that showed similar results would provide a stronger case, as would measurement of the interaction using an independent biophysical approach (since the authors reported purifying both of the proteins for structural analysis, this should be reasonably trivial).

a. The authors propose a model for how IsmP and ImcA putatively interact and a mechanism for how this interaction is regulated by iron binding. Beyond just being highly speculative, the data does not really seem to support these models except for the evidence for some role for iron regulating their, potentially, coordinated functions. While the B2H data may suggest an interaction, and the docking provides a possible binding pose, the structures themselves seem to indicate otherwise. The ImcA structure presented shows a relatively tight dimer that would have to dissociate to accommodate IsmP as proposed. No evidence is provided, nor is a means for how this would occur proposed.

Response: We agree with your suggestion to measure the interaction using an independent biophysical method. We will include these experiments in our revised manuscript. We have conducted Microscale Thermophoresis (MST) and Size Exclusion Chromatography (SEC) experiments (**Supplementary Fig. 2b**) to investigate the effect of ferric ions (Fe^{3+}) on the interaction between IsmP and ImcA. Our results indicate

that the binding of Fe³⁺ to IsmP leads to the dissociation of the IsmP-ImcA protein complex. Furthermore, MST data showed that IsmP^{E341A/A342R/R346A}, IsmP^{A342R}, IsmP^{E341A}, and IsmP^{L28R/G31R/W35R} did not bind to ImcA, which is consistent with our model. However, the exact interaction between ImcA and IsmP needs to be further investigated.

b. The proposed reorganization of IsmP upon iron binding is presumably based upon some observations of presumed flexibility (from B-factors and other observations), which are not compelling. The site directed mutagenesis experiments described on pg. 13 were perhaps suggestive but, to me, were not convincing evidence that this interaction occurs as proposed. More structurally conserved mutants that showed similar results would provide a stronger case, as would measurement of the interaction using an independent biophysical approach (since the authors reported purifying both of the proteins for structural analysis, this should be reasonably trivial).

Response: You are so appreciated for your comments and suggestions on the proposed potential hetero-tetramer model. On the one hand, we have to admit that the proposed hetero-tetramer model wasn't the only single possible candidate model. We already added some sentences to declare this point. On the other hand, whatever the model is, it should make little effect on our essential conclusions, which include but shouldn't be limited 1) ImcA is a DGC protein; 2) IsmP could interact with ImcA inhibiting DGC activity of the latter; 3) iron could bind to IsmP abolishing the inhibitory effect on ImcA DGC activity; 4) the level of ImcA DGC activity will control the biofilm level, motility... All above were used to support schematic shown in **Fig. 6**. And as for the interaction model (shown in **Fig. 6**) for IsmP and ImcA after they bind to each other, we also supplemented some details to indicate its uncertainty. Below is a much more detailed response about the proposed model:

First of all, what we want to emphasize is the hetero-dimer minimal unit must exist in the real IsmP+ImcA complex. Actually, the initial molecular docking was carried out using an ImcA monomer and an IsmP monomer. Certainly, what we got was a hetero-dimer (IsmP-ImcA) which was further verified by Biofilm assay (which suggested that IsmP mutant basically lost the inhibitory effect on ImcA (**Fig. 5e**)). Moreover, we supplemented some biophysical data from MST to further confirm the key residues and the existence of hetero-dimer interface (**Fig. 5f** and **Supplementary Fig. 2d**). In the initial IsmP-ImcA heterodimer model, the ligand binding pocket located in the DGC domain of ImcA faces toward the IsmP, which seems impossible to be realized for a ImcA homodimer without dissociation. Another thing is that due to the existence of detergent (the relative molecular weight of one single DDM micelle molecule is about 70 kDa (Strop, Pet al., *Protein Sci.*, 2005.)) and the instability for SEC method, it's a little difficult to determine the exact relative molecular weight of ImcA-IsmP polymer using SEC method (**Supplementary Fig. 2b**). Taken together, a possible minimal biological assembly unit for IsmP-ImcA complex was thereby proposed here.

3. In quite a few places throughout the manuscript, insufficient details were provided to be able to properly interpret the results including in the figure captions. Some

examples include: how the CHASE4 domain was identified/chosen and which residues were included for the protein expression and structure; how the mutagenesis experiments were designed and carried out (paragraph 2, page 12, in particular); many of the figure captions provide insufficient details including data presented in Fig. 2a, details about the maps shown in **Fig. 4**, and details about the gels shown in Supplementary Fig. 4 (what is in each lane), among others.

Response: Thank you for your insightful comments and suggestions. We have provided sufficient details in the Results and Methods parts.

4. Some of the experimental details are missing. Examples include how the CHASE4 domain protein was expressed and purified, how the proteins were expressed and treated for the Mn or Fe supplementation experiments (ie was it just the media that was supplemented and/or were the metal salts added to purified proteins?); native page data was reported (Supplementary Fig. 4) but not described etc.

Response: Thanks. We have added the experimental details in Methods section according to your suggestions.

Minor issues:

1. p2, l32: it is not clear what is meant by “that besides the catalytic sites”

Response: Sorry about for the confusion. We have rewritten this sentence in text (Line 31).

2. the authors need to define ImcA and IsmP earlier; ie perhaps include a description like what is given in lines 114-117 earlier in the text

Response: We have defined IsmP and ImcA following to the description in text.

3. this would benefit from a brief introduction to the assay design for what “wrinkled colonies” represent

Response: We have added a brief introduction for “wrinkled colonies (Lines: 102-103).

4. l202: the reference to Fig. 2d is incorrect

Response: We have changed Fig. 2d to Fig. 2a.

5. l260: “non-crystallographic” is not correct since this is not a crystal structure

Response: Thanks for the suggestion. We have corrected it in text.

6. l1304-305: shouldn't use “Obviously” and “presumably” together this way

Response: Thanks. We have replaced it with “It's rational to infer that...” (Line 305).

7. l378: the font appears to be different here

Response: We have corrected it.

8. l571: MnCl₂ and FeCl₃ are not metal ions

Response: We have changed MnCl_2 and FeCl_3 to Mn^{2+} and Fe^{3+} , respectively.

9. l643: should read “sharpened by Phenix_auto_sharpen”

Response: Thanks. We have changed it in text.

10. Supplementary Fig. 4 and 8. Captions – text at end is somehow different???

Response: We have changed the captions of Supplementary Fig. 4 and 9.

10. There is disagreement between the data shown in extended Table 2 and what is reported in the text (resolution and space group, in particular)

Response: Thanks. We have corrected it in the text.

11. The methods appear to describe the expression of full length IsmP, but the structure solved was only the CHASE domain

Response: We have provided the details for the expression and purification of CHASE domain in the Methods section (Lines: 494-501, 694-700).

12. It would be good to include the protein only control (eg. ImcA without GTP added) for the HPLC experiment, to see what may copurify

- Why are there multiple peaks for the GTP standard in HPLC?

Response: We have repeated the HPLC assays and included the protein only control (Fig. 2a). Multiple peaks for the GTP standard in HPLC maybe due to the conditions used in the previous experiments.

13. The representations of the GMPCPP binding to IMCA shown in Fig. 4 do not do a good job of showing how well the density corresponds to the putative ligand. I would suggest additional figures (perhaps in the Supplementary) showing difference/composite omit maps from a couple of different angles to provide convincing evidence of the fits.

Response: Thanks for your suggestions. We feel a little ashamed for having no idea of how to generate a difference/composite omit maps from a CryoEM volume file (ImcA is a Cryo-EM structure). We even consulted several professors who engaged in Cryo-EM, but still have no clue for how to make it. Thus, we choose to draw additional figures based on the original Cryo-EM map from several different angles to provide evidence of the fits (Supplementary Fig. 8)

14. Some of the text in figures is very small, particularly in the extended data figures. For example, the labeling of the ITC data in extended data Fig. 4.

Response: We have changed the figures to make larger.

Reviewer #2 (Remarks to the Author):

This manuscript by Zhan et. al. describes the discovery of a novel iron-sensing c-di-GMP signaling module consisting of two membrane proteins the authors designate IsmP,

which senses iron, and ImcA, a diguanylate cyclase (DGC) that synthesizes c-di-GMP. The authors provide evidence that these two proteins interact and at low concentrations of iron, IsmP inhibits the DGC activity of ImcA. Iron binding to the periplasmic CHASE4 domain of IsmP then causes it to dissociate with ImcA, leading to increase c-di-GMP and biofilm formation. I think the experiments are well done and clearly described, and the authors present significant in vivo and in vitro data to support this model. However, I do have some concerns about the physiological relevance of this iron responsive module as none of the relevant phenotypes are shown in a WT cell and the delta ismP mutant does not have a hyper-biofilm forming phenotype. I also think the authors oversell the evidence that this pathway exhibits signaling specificity. However, given the paucity of known environmental cues that regulate c-di-GMP signaling, and the mechanism by which they are sensed, this work will be of significant interest to the field and make a significant impact, so overall I am enthusiastic about it. Specific comments are:

Response: We sincerely thank the Reviewer's positive evaluation for our manuscript. The constructive comments and suggestions that have helped us improve the quantity of our manuscript.

1. One significant concern I have with the manuscript as I highlight below in points 3 and 7 is that the interesting phenotypes are only observed when *imcA* is overexpressed in the *ismP* deletion mutant. For example, in Fig. 2d. biofilms of the WT strain are not impacted by changes to iron. This brings into question the physiological relevancy of the results. The model in Fig. 6 then is not entirely accurate as such impacts were not shown in a WT cell. The authors should address this discrepancy in this discussion and argue for the relevance of the results.

Response: Thanks for the constructive suggestions. We have found that IsmP and ImcA interact with 10 and 6 c-di-GMP metabolic enzymes, respectively (**Supplementary Fig. 9b**), which is composed of a complex c-di-GMP regulatory networks. In order to determine the effect of iron and ImcA on biofilm formation via interacting with IsmP, we have tried to find out an appropriate concentration of Fe^{3+} for this experiment. We measured biofilm production of WT PAO1, $\Delta ismP/EV$, $\Delta ismP/p-ismP$, PAO1/*p-imcA* and $\Delta ismP/p-imcA$ in LB medium supplemented with different concentrations of Fe^{3+} . It is noted that biofilms of WT PAO1 overexpressing *imcA* are significantly increased but the WT strain is not impacted by changes to 100 $\mu\text{M Fe}^{3+}$ (**Supplementary Fig. 4d**), which suggests that Fe^{3+} plays a critical role in stimulating the DGC activity of ImcA. Importantly, our data showed that, like to PAO1/*p-imcA*, the biofilm production of WT PAO1/EV is also significantly increased in LB medium with 200 $\mu\text{M Fe}^{3+}$. However, no difference was observed for the $\Delta ismP/EV$ strain (**Supplementary Fig. 4d**), indicating that iron regulates biofilms via IsmP. Combination with our biochemical data, these findings pinpoint that binding of Fe^{3+} with IsmP alleviates the IsmP-ImcA interaction and then promotes the DGC activity of ImcA (**Fig. 2a**), which provides insight into how *P. aeruginosa* flexibly regulates intracellular c-di-GMP to control specific effector/target systems. We have discussed these issues in more detail in Discussion section.

2. The discussion at multiple points suggests that their results demonstrate the ImcA and IsmP module shows localized signaling specificity, but I would disagree with this statement as they provide no evidence for such localized signaling. They first make this claim in Lines 415-416 stating “Therefore, IsmP-ImcA module meets a local c-di-GMP signaling model.”, but this seems to be based on similarities to the NosP-NahK signal. In my view, their data suggests a generalized signaling module. In order to show localized or specific signaling, there needs to be a significant impact of the activity of a DGC or PDE on a c-di-GMP-regulated process in the absence of global changes to c-di-GMP. However, the author’s data, particularly in Fig. 1, show that biofilm formation and colony morphology changes are only apparent in strains where the overall concentration of c-di-GMP increases. If there are other pieces of evidence to suggest specific signaling, the authors need to do a better job of making their case. Otherwise, any claim for specific signaling should be removed.

Response: Thanks for the good comments. We agree with the reviewer that we do not provide sufficient evidence to meet a local signaling model. We have revised the title and deleted the “local” statement in text.

3. Extended Data 1: If IsmP is inhibiting the DGC activity of ImcA, then why does the delta *ismP* mutant exhibit reduced biofilm? In other words, why do the authors need to overexpress *IsmA* to see a phenotype?

Response: Thanks for the constructive suggestions. Our data show that overexpression of the DGC *ImcA* reduces biofilm production after strains cultured 14 hours (Supplementary Fig. 1) and no difference is observed after over 20 hours of cultures (Fig. 2d and Supplementary Fig. 3a)), which is consistent with previous reports (Bhasme *et al.*, *Microbiologyopen*, 2020; Kulesekara *et al.*, *PNAS*, 2006).

Based on this observation, we hypothesize that the biofilm phenotype of the *IsmP* mutant and the *ImcA*-overexpression strain may be influenced by a complex protein-protein interacting networks. We have identified 10 c-di-GMP metabolic enzymes interacting with *IsmP* and 6 with *ImcA*, respectively. We reason that the compensatory mechanisms may exist to maintain the balance of c-di-GMP via this complex network. Mutation of *ismP* does not affect *ImcA* alone, which may lead to a reduction of biofilm. Similarly, overexpression of *ImcA* also influences the function of several proteins. Therefore, we need to overexpress *imcA* in *ismP* mutant to evaluate the inhibitory effect of *IsmP* on *ImcA*.

4. Lines 96-117-This paragraph is a bit confusing as the authors switch between using PA2072 and *IsmP* without first defining that *IsmP* is encoded by this gene. I suggest they use the PA2072 designation until line 115 when they now define it as *IsmP*.

Response: Thanks. The PA2072 designation is used until line 116. We have revised it in text and figures.

5. Line 137-This should be Extended Data Fig. 3, and I do not see any data to show wrinkling on CR plates.

Response: Sorry about for this mistake. We have added the missing data showing wrinkling on CR plates in Supplementary Fig. 3a.

6. Line 142-Again, there is no wrinkling colony morphology in Extended Data Fig. 3. I think the authors are referring to Fig. 1d.

Response: We have added the corresponding CR plates to Supplementary Fig. 3a.

7. Fig. 1e, f-Similar to point 3 above, why doesn't the delta *ismP* mutant exhibit enhanced c-di-GMP levels as *imcA* would be derepressed. Or is *IsmA* not expressed in this mutant?

Response: Please see the comment 3. In addition, we show that protein level of *ImcA* does not significantly change in the *ismP* mutant compared to wild-type parent (Supplementary Fig. 9d), which further supports the complex regulatory network of the c-di-GMP signaling.

8. Line 188-As K_d refers to the concentration at which half of the protein is bound to the small molecule and is the inverse of the association constant, PA0847 has a weaker binding affinity for Fe^{+3} at 16 μM compared with *IsmP* CHASE domain at 6.9 μM .

Response: Thanks. We have revised it as "PA0847 has a weaker binding affinity for Fe^{+3} at 16 μM compared with the *IsmP* CHASE domain at 6.9 μM ".

9. Fig. 2d-This might be my pdf file, but the text on this figure is not sharp and difficult to read. It is also worth highlighting what conditions were used for this experiment. For example, the authors should state that BIP is an iron chelator.

Response: We have enlarged the text in Fig. 2d and provided the detail conditions used for this experiment.

10. Line 456-The authors should reference the plasmids and primers table in this section.

Response: We have referenced the plasmids and primers table in line 493.

Reviewer #3 (Remarks to the Author):

The authors identify PA2072 as a gene that, when either overexpressed or deleted, led to a decreased biofilm production. The authors performed bacterial two hybrid screens and identified a number of proteins involved in c-di-GMP synthesis and degradation that interact with PA2072 including PA1851. The authors show that overexpression of PA1851 (renamed *imcA*) in a strain lacking PA2072 (renamed *ismP*) led to increased biofilm. The author perform biochemistry and show that *ImcA* can synthesize c-di-GMP in vitro, but this activity is inhibited by the addition of *IsmP* or the *IsmP* Chase4-PASPAC domain. The inhibition by *IsmP* can be reversed by the addition of Fe^{3+} . The authors solved the crystal structure of the CHASE4 domain of *IsmP* and modeled in the Fe^{3+} . Alteration of the predicted residues that bind Fe^{3+} prevented binding to iron and addition of Fe^{3+} no longer reversed inhibition of *ImcA*. The authors solved the EM structure of *ImcA* with a non-hydrolysable GTP analog, which was similar to the AlphaFold model. The authors modeled *IsmP* and *ImcA* which led to a number of

residues predicted to mediate the interaction. Site directed mutants of those residues prevented inhibition of ImcA diguanylate cyclase activity by IsmP.

Response: We sincerely thank the Reviewer's positive evaluation for our manuscript. The constructive comments and suggestions that have helped us improve the quantity of our manuscript.

Major comments:

1. The pull-down is not very convincing. If the authors have the purified proteins, can the be mixed and analyzed by SEC as shown in Extended Data Fig. 4F. This would provide better support for the interaction between these two proteins than the docking model. This should also be done with Fe^{3+} to show the complex dissociates.

Response: Thanks for the good suggestions. We have conducted SEC analysis of IsmP-ImcA interaction and showed that Fe^{3+} causes IsmP_{His}-ImcA complex to dissociate. Additionally, this observation is confirmed by MST detection (**Supplementary Fig. 2b**).

2. The authors' model suggest that inhibition is based on stoichiometric interaction between IsmP and ImcA. This raises a number of questions. Shouldn't overexpression of ImcA overcome inhibition by IsmP, but this is not the case in Figure 2D (PAO1/EV vs PAO1/p-*imcA* in LB). How much IsmP protein is in PAO1 strain? How much ImcA is made when expressed from the plasmid?

Response: Thanks for the constructive suggestions. Please see the comments 3 and 7 of reviewer 2. Moreover, we tested the protein levels of ImcA-eGFP and IsmP-Flag in PAO1/ Mini-CTX-*ismP*-Flag-*imcA*-eGFP, PAO1/p-*imcA*-eGFP/ Mini-CTX-*ismP*-Flag, and Δ *ismP*/p-*imcA*-eGFP strains. The Mini-CTX plasmid is integrated into the chromosome of these strains. We found that the mutation of *ismP* does not affect the expression of *imcA*, and protein levels of *ismP* and *imcA* in WT PAO1 are similar. This result validates our hypothesis that the IsmP-ImcA module with the other 9 c-di-GMP metabolic proteins can flexibly maintain the homeostasis of intracellular c-di-GMP.

3. Figure 2D shows that addition of iron to PAO1/EV does not have any effect on biomass of the biofilm. This raises a follow-up to the above point is how much ImcA is in PAO1? Is it just not expressed?

Response: Thanks for the constructive suggestions. We have found that IsmP and ImcA interact with 10 and 6 c-di-GMP metabolic enzymes, respectively (**Supplementary Fig. 9b**), which is composed of a complex c-di-GMP regulatory networks. In order to determine the effect of iron and ImcA on biofilm formation via interacting with IsmP, we have tried to find out an appropriate concentration of Fe^{3+} for this experiment. We measured biofilm production of WT PAO1, Δ *ismP*/EV, Δ *ismP*/p-*ismP*, PAO1/p-*imcA* and Δ *ismP*/p-*imcA* in LB medium supplemented with different concentrations of Fe^{3+} . It is noted that biofilms of WT PAO1 overexpressing *imcA* are significantly increased but the WT strain is not impacted by changes to 100 μM Fe^{3+} (**Supplementary Fig. 4d**), which suggests that Fe^{3+} plays a critical role in stimulating the DGC activity of ImcA. Importantly, our data showed that, like to PAO1/p-*imcA*, the biofilm production of WT

PAO1/EV is also significantly increased in LB medium with 200 μM Fe^{3+} . However, no difference was observed for the $\Delta\text{ismP}/\text{EV}$ strain (**Supplementary Fig. 4d**), indicating that iron regulates biofilms via IsmP. Combination with our biochemical data, these findings pinpoint that binding of Fe^{3+} with IsmP alleviates the IsmP-ImcA interaction and then promotes the DGC activity of ImcA (**Fig. 2a**), which provides insight into how *P. aeruginosa* flexibly regulates intracellular c-di-GMP to control specific effector/target systems. We have discussed these issues in more detail in Discussion section.

4. The authors propose Fe^{3+} binding in the micromolar range. The authors should discuss when is amount of Fe^{3+} likely to be possible in the environment or during infection.

Response: We have discussed it in text (Lines: 379-384).

5. While a writing issue, this is a major problem since the authors starts using IsmP (line 100) and ImcA (line 110) prior to naming the two proteins in line 115-117. Please introduce the name before using it. Prior to the renaming, just use the PA gene number. Please stop using the PA gene number in Material and Methods and Extended Data since you are renaming them.

Response: Thanks. We have changed it in text and figures.

Minor issues:

1. For the bacterial two hybrid assays, showing the bait plasmid with an empty prey plasmid would be a better negative control than two empty vectors.

Response: We used the bait plasmid with an empty prey plasmid as a negative control in **Fig.1c** and **Supplementary Fig. 2c**.

2. Abstract – several statements are overgeneralizations. Please be more specific. Line 23 – “However, environmental signals controlling the intracellular c-di-GMP levels still remain enigmatic.” and Line 36 “CHASE4 domain directly senses the environmental signals” The authors should specify iron since many environmental signals are known for enzymes regulating c-di-GMP.

Response: We have revised the statements in text.

3.Line 32 – please clarify meaning of “unveiled a unique conformation that besides the catalytic site”

Response: We have reorganized this sentence (line 31).

4. Replace reference #5 in line 50. This is not an appropriate reference for this statement.
Response: Thanks. We had replaced this reference with a more suitable one that accurately supports the statement.

5. Add reference for PdeR-DgcM-MlrA in line 63.

Response: We have added it.

6. The biochemical data in Figure 2A needs to be clarified and quantified.

A. The amount of proteins used in diguanylate cyclase assay is not mentioned for Fig 2A. This makes it nearly impossible to understand the data being presented.

Response: We have provided the concentrations of proteins, GTP, and c-di-GMP in Fig.2a.

B. Different ratios of the ImcA and IsmP should be used to show that there is a stoichiometry for the observed inhibition.

Response: We have added the protein ratios of ImcA and IsmP in Fig. 2a.

C. The authors states in line 165 that “However, GTP remained the primary compound in the reaction...” The figure shows that c-di-GMP is being made, so quantification of the data is needed.

Response: We have added the relative quantification of c-di-GMP in Fig 2a.

7. Authors should test IsmP Chase4-PASPAC to inhibit Δ ismP/p-imcA. This data should go into Figure 3I.

Response: We have tested it and added the result in Figure 3i.

8. Avoid using the word – obvious or obviously. Let the readers make that determination.

Response: Thanks. We have deleted “obviously” (lines 305, 324 and 419) in these sentences.

9. The title is “A local signaling switch controls *Pseudomonas aeruginosa* behaviors in response to iron”. Why is IsmP and ImcA localized signaling? Is it just because they physically interact? Local signaling suggest that the signaling molecule is signaling locally rather than globally, so the system described doesn’t seem to show that. Perhaps change title to “A c-di-GMP signaling module controls *Pseudomonas aeruginosa* behaviors in response to iron”. This is more accurate to the proposed model.

Response: We greatly appreciate your valuable suggestion on the title. we have changed it to “A c-di-GMP signaling module controls *Pseudomonas aeruginosa* behaviors in response to iron”. Additionally, we have removed the definition of local signaling transduction of IsmP and ImcA in the discussion section (lines 448-465).

Reviewer #1 (Remarks to the Author):

Overall, the authors reasonably addressed the majority of the points that were raised. I have a few other relatively minor points that I believe should be addressed regarding some of the statements and the ITC data. Also note that there is a fair amount of incorrect or poor language usage, but I assume that this will be addressed during editing.

With these points addressed, I feel that it would be appropriate to be published.

In the revised text, pg 13, ll 335-336, the authors state that "...a ImcA-IsmP hetero-dimer must be found..." As the authors state themselves, their methods cannot conclusively determine the oligomeric form. I would suggest changing the wording here to say, for example, that an "ImcA-IsmP hetero-oligomer is present" or an "ImcA-IsmP complex is present". While it may be likely that a hetero-dimer is present, without definitive evidence of the interactions or stoichiometry, I think it best to be careful with the wording here.

While the author's response in the rebuttal letter about the Fe coordination site are reasonable, I would suggest some of these statements appear in the text (pg 9, ll 228-236, it appears that no changes were made in the revised manuscript). Specifically, the authors state that they were unable to crystallize with Fe present. Considering all their other data, it would be warranted to speculate why these experiments didn't work (although this is a very minor point). More importantly, I feel that it should more clearly be stated in the text or the figure caption (Fig. 3) that this experiment was used to identify the putative iron binding site and that the modeled structure at that site is a rough model and doesn't have the correct geometry or bond distances. Regarding the latter point, the bond distances should likely be removed from Figure 3g, since some of these distances are likely too large (4.1 and 4.5 Å do not really make sense as appropriate bond distances here), and since this is modeled and the distances are expected to be inaccurate/approximate.

The values reported on page 8, l 187 and page 15, ll 187, 386-387, derived from the fits to the ITC data given in the manuscript are clearly not accurate values. The curve fits (Figs. 2c and S4e) do not fit the data well (quite poorly for Fig. S4e). Either the curve fits need to be recalculated in order to derive accurate values that can be reported, or the reported values need to be qualified as approximate or, perhaps, given as a range. Certainly, the level of accuracy reported (reported to one decimal point) is not appropriate given the poor fits to the data.

Reviewer #2 (Remarks to the Author):

This revised manuscript by Zhan et. al. somewhat addressed my two major concerns which were no evidence for a lack of signaling specificity and no response of the WT signaling system to iron, which had previously been only demonstrated in the delta ismP mutant overexpressing icmA. In regards to the first point, the authors no longer claim there is signaling specificity. In regards to the second point, the authors now show that iron addition does enhance biofilm formation in the WT strain but not in the delta ismP mutant. I am somewhat concerned by the high concentration of iron that was required to see this effect though. The authors argue that perhaps the multiple interactions of these two proteins with other components of the c-di-GMP signaling system mask effects in the WT strain, which is a fair point. Thus, given the significant other data presented, I am willing to accept their model as supported, but clearly more work needs to be done to understand the role of these proteins in their native context. Besides these two points, I did notice a few other minor things as listed below.

1. Line 84-As one of the reviewers pointed out, the authors do not demonstrate direct binding of IsmP to heme.
2. Fig. S1-I don't see any data regarding overexpression of PA2072.
3. Fig. 2b-I don't understand why the low iron condition would inhibit the growth of strains lacking ismP. Are the authors suggesting that IsmP inhibition of IcmA in low iron is important for robust growth? If so, then does deletion of icmA in the ismP mutant suppress this growth inhibition? A

little more context is needed for this result.

Reviewer #3 (Remarks to the Author):

Thanks for addressing the comments in a thoughtful manner.

Dear editor and reviewers:

Thank you for your letter and for the reviewers' comments concerning our manuscript entitled "A c-di-GMP signaling module controls *Pseudomonas aeruginosa* behaviors in response to iron" (ID: NCOMMS-23-51914A). Herein we have carefully addressed concerns and suggestions raised by the editor and reviewers and provide our point-by-point responses below.

Reviewer #1 (Remarks to the Author):

Overall, the authors reasonably addressed the majority of the points that were raised. I have a few other relatively minor points that I believe should be addressed regarding some of the statements and the ITC data. Also note that there is a fair amount of incorrect or poor language usage, but I assume that this will be addressed during editing. With these points addressed, I feel that it would be appropriate to be published. **Response: We sincerely thank the Reviewer's positive evaluation for our manuscript. The constructive comments and suggestions that have helped us improve the the quality of our manuscript. We provided the point-by-point responses below.**

In the revised text, pg 13, ll 335-336, the authors state that "...a ImcA-IsmP hetero-dimer must be found..." As the authors state themselves, their methods cannot conclusively determine the oligomeric form. I would suggest changing the wording here to say, for example, that an "ImcA-IsmP hetero-oligomer is present" or an "ImcA-IsmP complex is present". While it may be likely that a hetero-dimer is present, without definitive evidence of the interactions or stoichiometry, I think it best to be careful with the wording here.

Response: Thanks. We have changed the statement to "ImcA-IsmP hetero-oligomer is present" in text.

While the author's response in the rebuttal letter about the Fe coordination site are reasonable, I would suggest some of these statements appear in the text (pg 9, ll 228-236, it appears that no changes were made in the revised manuscript). Specifically, the authors state that they were unable to crystallize with Fe present. Considering all their other data, it would be warranted to speculate why these experiments didn't work (although this is a very minor point). More importantly, I feel that it should more clearly be stated in the text or the figure caption (Fig. 3) that this experiment was used to identify the putative iron binding site and that the modeled structure at that site is a rough model and doesn't have the correct geometry or bond distances. Regarding the latter point, the bond distances should likely be removed from Figure 3g, since some of these distances are likely too large (4.1 and 4.5 Å do not really make sense as appropriate bond distances here), and since this is modeled and the distances are expected to be inaccurate/approximate.

Response: Thanks for your valuable comments and suggestions.

1) Additional statements have been made in the figure caption to highlight the roughness of proposed model in our study. And minor modification was also made to corresponding sentence (page 9 || line 232-233); 2) Dash lines indicating bond distances

were already removed from Fig. 3g.

The values reported on page 8, l 187 and page 15, ll 187, 386-387, derived from the fits to the ITC data given in the manuscript are clearly not accurate values. The curve fits (Figs. 2c and S4e) do not fit the data well (quite poorly for Fig. S4e). Either the curve fits need to be recalculated in order to derive accurate values that can be reported, or the reported values need to be qualified as approximate or, perhaps, given as a range. Certainly, the level of accuracy reported (reported to one decimal point) is not appropriate given the poor fits to the data.

Response: Thanks. The reported values in Figs. 2c and S4e are qualified as approximate and data are given as a range.

Reviewer #2 (Remarks to the Author):

This revised manuscript by Zhan et. al. somewhat addressed my two major concerns which were no evidence for a lack of signaling specificity and no response of the WT signaling system to iron, which had previously been only demonstrated in the delta ismP mutant overexpressing icmA. In regards to the first point, the authors no longer claim there is signaling specificity. In regards to the second point, the authors now show that iron addition does enhance biofilm formation in the WT strain but not in the delta ismP mutant. I am somewhat concerned by the high concentration of iron that was required to see this effect though. The authors argue that perhaps the multiple interactions of these two proteins with other components of the c-di-GMP signaling system mask effects in the WT strain, which is a fair point. Thus, given the significant other data presented, I am willing to accept their model as supported, but clearly more work needs to be done to understand the role of these proteins in their native context. Besides these two points, I did notice a few other minor things as listed below.

Response: Thank you very much for your valuable comments and suggestions. We understand that you concern about the high concentration of iron required to observe this effect, but the biofilm results are based on the repeated experiments in the present conditions. We appreciate your acceptance of our model, and more work needs to be done to understand the role of these proteins in their native context in the future.

1. Line 84-As one of the reviewers pointed out, the authors do not demonstrate direct binding of IsmP to heme.

Response: We agree with the reviewer that heme binding data is incomplete. The heme reagent can only be dissolved with a strong alkali. We attempted to adjust the pH value but was unsuccessful due to the high background heat, which precludes the ITC experiment for testing the interaction between heme and CHASE4. Therefore, we only measured the effects of heme on bacterial growth and biofilm phenotype.

2. Fig. S1-I don't see any data regarding overexpression of PA2072.

Response: Sorry about for this mistake. The second of PA1851 should be PA2072, and we have corrected it in this figure (**Supplementary Fig.1**).

3. Fig. 2b-I don't understand why the low iron condition would inhibit the growth of strains lacking *ismP*. Are the authors suggesting that *IsmP* inhibition of *IcmA* in low iron is important for robust growth? If so, then does deletion of *icmA* in the *ismP* mutant suppress this growth inhibition? A little more context is needed for this result.

Response: Thanks for the good comment. In our previous experiments, we compared the growth of $\Delta ismP$ with $\Delta ismP\Delta icmA$ double mutant strain under low iron conditions. Data show that the growth rate of $\Delta ismP$ is the same with $\Delta ismP\Delta icmA$ strain, suggesting that the defective growth of $\Delta ismP$ in low iron may be due to *IsmP* affects other metabolic pathways. The detail mechanism needs to be further investigated.

Reviewer #3 (Remarks to the Author):

Thanks for addressing the comments in a thoughtful manner.

Response: Thanks.